# Machine learning-coupled combinatorial mutagenesis enables resource-efficient engineering of CRISPR-Cas9 genome editor activities

Dawn G. L. Thean[1,9], Hoi Yee Chu [1,2,9], John H. C. Fong[1], Becky K. C. Chan [1,2], Peng Zhou[1,3], Cynthia C. S. Kwok[1], Yee Man Chan[4], Silvia Y. L. Mak [4], Gigi C. G. Choi [1,2], Joshua W. K. Ho [5,6], Zongli Zheng [4,7,8] & Alan S. L. Wong [1,2,3✉]

The genome-editing Cas9 protein uses multiple amino-acid residues to bind the target DNA. Considering only the residues in proximity to the target DNA as potential sites to optimise Cas9's activity, the number of combinatorial variants to screen through is too massive for a wet-lab experiment. Here we generate and cross-validate ten in silico and experimental datasets of multi-domain combinatorial mutagenesis libraries for Cas9 engineering, and demonstrate that a machine learning-coupled engineering approach reduces the experimental screening burden by as high as 95% while enriching top-performing variants by ∼7.5-fold in comparison to the null model. Using this approach and followed by structure-guided engineering, we identify the N888R/A889Q variant conferring increased editing activity on the protospacer adjacent motif-relaxed KKH variant of Cas9 nuclease from *Staphylococcus aureus* (KKH-SaCas9) and its derived base editor in human cells. Our work validates a readily applicable workflow to enable resource-efficient high-throughput engineering of genome editor's activity.

[1] Laboratory of Combinatorial Genetics and Synthetic Biology, School of Biomedical Sciences, The University of Hong Kong, Hong Kong, SAR, China. [2] Centre for Oncology and Immunology Limited, Hong Kong Science Park, Hong Kong, SAR, China. [3] Department of Electrical and Electronic Engineering, The University of Hong Kong, Hong Kong, SAR, China. [4] Ming Wai Lau Centre for Reparative Medicine, Karolinska Institutet, Hong Kong, SAR, China. [5] School of Biomedical Sciences, The University of Hong Kong, Hong Kong, SAR, China. [6] Laboratory of Data Discovery for Health Limited (D24H), Hong Kong Science Park, Hong Kong, SAR, China. [7] Department of Biomedical Sciences, City University of Hong Kong, Hong Kong, SAR, China. [8] Biotechnology and Health Centre, City University of Hong Kong Shenzhen Research Institute, Shenzhen, China. [9] These authors contributed equally: Dawn G. L. Thean, Hoi Yee Chu. ✉email: aslw@hku.hk

The CRISPR-associated protein-9 (Cas9) protein has become an important tool for genome editing. The specificity of Cas9 is guided by a single-guide RNA (sgRNA) to the matching complementary genomic sites with a protospacer-adjacent motif (PAM). *Staphylococcus pyrogenes* Cas9 (SpCas9) is popularly used for genome editing due to its high editing efficiency at most targeting sites and short 5′-NGG-3′ PAM with less sequence constraint for editing more sites. This comes with concerns regarding the greater off-target effect of the nuclease that dampens editing accuracy. Multiple studies have been conducted to modify SpCas9 to optimize editing accuracy and reduce constraints for PAM recognition[1–10]. However, the bulkiness of SpCas9 limits its applications for in vivo genome editing as adeno-associated virus, the delivery tool for clinical gene therapy, has a cargo-packaging limit of ~4.4kb. Researchers have turned to small Cas9 orthologs with comparable activity to SpCas9, such as the *Staphylococcus aureus* Cas9 (SaCas9)[11]. Although SaCas9 is desirable in packaging for genetic therapeutics, its drawbacks are its longer PAM 5′-NNGRRT-3′, reducing genome coverage with room for improvement in gaining higher activity and specificity.

Thus far, most of the optimized Cas9 variants possess 2–7 mutations scattering over multiple domains of the protein[1–9,12–16]. Interestingly, each of the unique mutation combinations has contributed to comparable performance and editing fidelity. For example, >30 and >19 different amino-acid sites are engineered among the >13 SpCas9 and >10 SaCas9 variants, respectively. Still, they represent only a small proportion of amino-acid sites interacting with the sgRNA–DNA complex[17,18], each site is a potential candidate for optimization. A systematic experimental screen across all the candidate amino-acid positions to identify the best-performing Cas9 variants will be prohibitively labor-intensive and expensive. Therefore, here we explored the use of a machine-learning (ML)-guided approach to tackle the Cas9 optimization problem.

Machine learning is employed in a wide range of protein engineering tasks. In silico screens show great success in identifying high-performance variants of enzymes[19], ion channels[20,21], binders[22], and viral capsids[23]. In particular, the ML-based approach in the antibody maturation and viral capsid diversification involved fully saturated mutagenesis from 9 to 28 amino-acid sites. The capacity to evaluate such a large number of variants far exceeds what is feasible experimentally, even in massively parallel experiments. The main attracting reason of using ML is to reduce the burden of the experimental screen and narrow down top-performing candidates for further characterization and engineering. Studies have shown that the ML approach can reliably predict the fitness of the full virtual library ($10^5$–$10^{12}$ variants) using a small subsample of empirical fitness data ($10^3$–$10^4$ variants or even less)[22,24]. To minimize screening efforts, ML-guided approaches such as machine-learning-assisted approach to directed evolution (MLDE)[25,26] extrapolate from the experimentally determined fitness of a small sample of variants from a combinatorial mutant library to predict the full variant space covered by the multisite saturation mutagenesis library in silico. ML is compatible with screening platforms, which use fluorescence-activated cell sorting and next-generation sequencing as readouts, to evaluate the functionality of protein variants in a pooled library setting.

Here we describe how ML can be applied to a focused library derived from the structure-guided design. Such a focused library usually targets multiple sites (e.g., eight sites in our previous SpCas9 optimization[8]) key to the protein functionality with deliberated mutations that are restricted to a few residues per site. We show that ML-based in silico screens are efficient and accurate in independent Cas9 optimization tasks, reducing the wet-lab labor by as high as 95%. In this study, we aim to boost the activity

of SaCas9 while maintaining the broader PAM specificity. No mutation has been reported in SaCas9 that increases its editing activity before this work. We chose to base our modifications on the E782K/N968K/R1015H SaCas9 variant (KKH-SaCas9), which showed comparable activity with wild-type SaCas9 and recognized an expanded PAM 5′-NNNRRT-3′[13]. A prior study emphasized modifications of the PI domain in modifying around the PAM duplex region[14], however, no studies have looked at modifying the WED domain, which is responsible for the sgRNA scaffold recognition, of SaCas9 thus far. Combining ML-based and combinatorial mutagenesis screens with downstream structure-guided rational design and wet-lab validations, we uncover insights to changes in the WED domain that could provide stronger interactions with the PI domain, thereby increasing the DNA-binding ability of KKH-SaCas9 protein. Our results revealed that the modification on the WED domain may come through more often in enhancing the protein's activity rather than the changes in the PI domain. In addition, we tested the same mutation with a high-fidelity SaCas9 variant (KKH-SaCas9-SAV2) and a KKH-SaCas9-derived cytosine base editor, demonstrating that the mutations could be widely applicable to increase editing activity. This work also sets up a useful workflow and establishes parameters that can maximize ML usefulness in succeeding screens and minimize wasted wet-lab resources for engineering other components of the Cas9 system and gene editing tools.

## Results

**Validating MLDE model for predicting SpCas9's activity.** The vast combinatorial mutational space is an obvious challenge for protein engineering. Utilizing machine-learning-based methods empowers us to efficiently explore the functional impact brought by mutations and break through the experimental limits of testing more combinatorial mutants. Here we tested whether the ML-based in silico screen could be applicable on the Cas9 optimization problem, using a small fraction of variants with experimentally determined activities from a combinatorial mutant library. Specifically, using our previously published combinatorial mutagenesis data on SpCas9[8], we sought to find out the minimal sample size sufficient to accurately predict which variants possess top enzyme activities for the library. We used the MLDE package that predicts activities of variants from multi-site-saturated mutagenesis libraries from a small sample of variants. The MLDE packages offer numerous embedding and model parameters. We selected the simple Georgiev embedding[27] and the learnt embedding from Bepler et al.[28] combined with more complex neural network models (parameter 1) or with an ensemble of more simple models such as random forests and SVM (parameter 2) to model the activities of SpCas9.

We tested different input sizes (including 5%, 10%, 20%, 50%, and 70% of randomly downsampled empirical data points) as the training data for SpCas9 activity. We also explored whether using a sample with higher diversity (see "Methods" on variant selection) increases accuracy because a previous study showed that sampling diverse samples improves the ML performance[29]. Deciding on which characteristic of the data is most useful as the training data will help guide the library design for building variants for empirical testing. To this end, we selected more dissimilar variants by restricting the numbers of variants with merely one and two mutations apart to be included in the input dataset. Especially when there were few input data points (5%, 10%, and 20%), we observed that this selection scheme boosted the number of variants harboring three, seven, and eight mismatches from each other in the dataset (maximal increase by 29.1, 8.4, and 0.5% for Sg5 on-target activity dataset and 32.2,

9.89, and 12% for Sg8 on-target dataset, respectively) (Supplementary Fig. 1; Supplementary Data 1). When the sample size was increased to 50% empirical data, such a selection scheme did not confer more dissimilarities among variants compared with random selection, and thus we were only able to evaluate the effect of increased sequence diversity with samples with smaller sizes. We ran MLDE on all the datasets and calculated the precision, specificity, and sensitivity on predicting variants with at least 70% of wild-type activity. To ensure the prediction outcome was not affected by overlapping of training and testing data, 20% of variants in the library from the experimental dataset were randomly subset and withheld a priori, and had never been fed to the MLDE algorithm. Regardless of the little increase in diversity described above, the diverse dataset ended up with similar precision based on the four tested embedding and model parameters over the Sg5 dataset, compared with randomized selection (Supplementary Fig. 2; Supplementary Data 2). Upon closer examination, predictions from diverse and randomized training data led to similar results (Supplementary Fig. 3), indicating again the limited benefits gained from increasing sequence diversity in the training data. We used the randomized selection scheme for the subsequent protein-optimization problem.

We also evaluated the MLDE performance with enrichment score and normalized discounted cumulative gain (NDCG), which reflects the likelihood of identifying top-performing variants (see Methods). In our ML runs, we found that NDCG and enrichment score were robust metrics for scoring models and parameter performances (Supplementary Fig. 2), especially for Sg8 on-target activity where only about 10 variants (1.15% of the library) show activities comparable to WT[8]. NDCG and enrichment score were thus used for subsequent scoring with our objectives to isolate the top-performing Cas9 variants. Looking into NDCG and enrichment score, all the embeddings and model combinations performed well, while Bepler and Georgiev embeddings with p2 parameter outperformed other parameters when 5–20% of training data was fed to MLDE (Fig. 1a).

Taking NDCG and enrichment score together into consideration, we determined that 20% of input can be used as the input threshold that gave a robust and consistent performance in identifying top-performing candidates (Fig. 1a, b; Supplementary Fig. 4). About 10% of input can also be used to further reduce the experimental screening burden with the metric scores slightly compromised (Fig. 1a). Using merely 10% of input was sufficient to identify clusters of variants with high activity for the Sg5 dataset, and consistent identification of variants with at least 70% of wild-type activity across 10%, 20%, 50%, and 70% of input was observed (Supplementary Fig. 3). MLDE runs on the Sg8 dataset again successfully identified the top-performing variants (Fig. 1b, Supplementary Fig. 5), albeit that NDCG and enrichment score were lower than those observed for Sg5 dataset (Fig. 1a; see Supplementary Text). The top hits predicted from the Sg5 and Sg8 datasets included Opti-SpCas9 that was experimentally confirmed in our previous study to exhibit high on-target activities for both Sg5 and Sg8[8]. Using MLDE, the enrichment in identifying top-performing variants reached about 8.6-fold for Sg5 (and about 5.8-fold for Sg8) with 20% input compared with the null model (Fig. 1a; Supplementary Data 2). The enrichment reached about 7.5-fold with 5 and 10% input for Sg5 (Fig. 1a; Supplementary Data 2). We further applied MLDE for off-target prediction. We took the same set of variants used for on-target activity prediction constituting 10, 20, 50, and 70% of empirical data of Sg5 off-target activities as training data for MLDE. MLDE achieved similarly high NDCG scores and about 5.5-fold enrichment with 20% input in off-target activity prediction (Supplementary Fig. 6; Supplementary Data 3).

PAM relaxation is another key research area on SpCas9 engineering, and thus, we explored whether MLDE could facilitate screening on variants that cleave effectively on noncanonical PAMs. Specifically, we tested MLDE on SpCas9 variants' activities on noncanonical NGN PAMs from the previously published High-Throughput PAM Determination Assay (HT-PAMDA) experiment[30]. We run MLDE using 10, 20, 25, and 50% input (6, 12, 15, and 29 variants). Due to the small size of the library, we were not able to calculate enrichment (see Methods) since there were only 3 variants warranted to be top 5% in the dataset. We focused on NDCG and again observed high scores on MLDE's prediction (Fig. 1c; Supplementary Data 4). All the modeling parameters performed well when supplied with 50% training data, while Bepler and Georgiev embeddings with p2 parameter outperformed other parameters when only 10 and 20% training data were fed to MLDE (Fig. 1c). Across the four PAMs tested, supplying 20% training data to MLDE could achieve comparable performance to MLDE runs using 25 and 50% training data. Thus, we used MLDE results from 20% training data for the rest of the analysis. Looking into the best runs for each PAM, SpG was detected correctly to be among the top 20% variants with high activity at NGAT and NGCC PAMs (Fig. 1d; Supplementary Fig. 7).

Taking together the accurate prediction and the ability to isolate bona fide high-activity variants, we found that MLDE is compatible with rational-design-guided library in various aspects of SpCas9 engineering.

**Experimentally validated MLDE prediction identifies activity-enhanced KKH-SaCas9 variants.** Using the parameters that yielded a good prediction of SpCas9's activity, we attempted to apply MLDE to tackle the SaCas9 optimization problem. We sought to augment the editing activity of KKH-SaCas9 and speculated that introducing additional non-base-specific interactions between KKH-SaCas9 and the PAM duplex of the target DNA could increase the enzyme's efficiency. Such a strategy was shown to be effective in compensating the reduced DNA base-specific interactions and restoring the activity of an engineered SpCas9 variant with broadened PAM compatibility[4]. For SaCas9, Nishimasu et al. have illustrated in the crystal structure (5CZZ) its amino-acid residues that show direct contact with the target DNA of the PAM duplex[18]. Specifically, it was highlighted that amino-acid residues at position 985, 986, 991, and 1015 on its PI domain form water-mediated hydrogen bonds with the nontarget DNA strand at the PAM duplex, while residues at positions 789, 882, 886, 887, 888, 889, and 909 on its WED domain interact with the phosphate backbone of the PAM duplex. Mutations at positions 988 and 989 were also reported to alter SaCas9's PAM constraint[14]. In this study, we focused on modifying eight amino-acid positions (887, 888, 889, 985, 986, 988, 989, and 991) that interact with and surround the PAM duplex for combinatorial mutagenesis (Fig. 2a; Supplementary Data 5). Up to two amino-acid alternatives to the wild-type residue were selected for each site based on structural predictions. This could potentially increase non-base-specific interactions between KKH-SaCas9 and the DNA and relieve the PAM constraint. To facilitate the changes, we selectively chose sites in the WED domain to reinforce the protein binding to the DNA backbone (Supplementary Data 5). This led to a total of 1,296 variant combinations, including the wild-type residues (i.e., 12 mutation combinations at WED domain x 108 mutation combinations at PI domain). We did not modify residue position 1015 because this R1015H was shown to be important for maintaining the high activity of KKH-SaCas9 to act on NNNRRT PAM[13]. Residue positions 789, 882, 886, and 909 were not included to confine the library size for

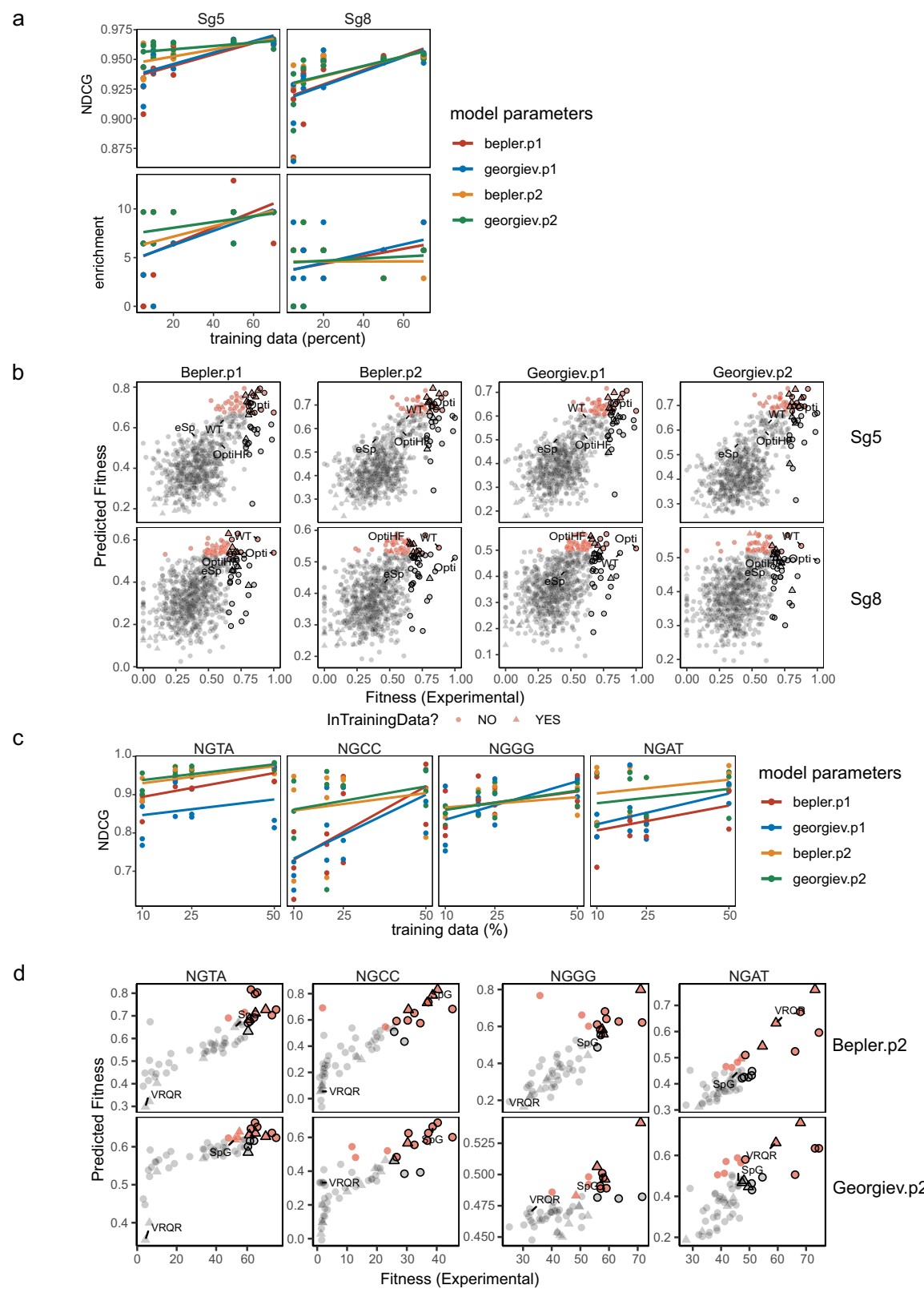

combinatorial mutagenesis, while they are potential sites for future engineering.

We randomly picked 260 out of the 1,296 (20%) variants, generated empirical data from a screening library (see below) as training set input, and run MLDE with combinations of Bepler or Georgiev embeddings and modeling parameters 1 or 2 to predict

functional variants that have comparable activities to wild-type (at least 70%) from the full variant space. Parallelly, we sought to confirm our in silico prediction results with the experimental screening data to validate the MLDE model that predicts KKH-SaCas9's activity with high accuracy. We assembled a full-coverage screening library of 1296 variants, and the library was

**Fig. 1 Performance of MLDE predictions on SpCas9's activities. a** Performance of MLDE predictions on Sg5 and Sg8 on-target activities with SpCas9. Enrichment score and NDCG of MLDE runs using a combination of embeddings (Bepler/Georgiev) and models (p1 and p2) are plotted against the size of randomly selected training data. The best-fit line summarizes 3 replicates for each embedding and model parameter combination of MLDE runs using 5, 10, 20, 50, and 70% of training data. **b** Predicted versus empirical fitness of variants in the best-performing MLDE runs using 20% input training data, given the different combinations of embeddings and model parameters. The predicted fitness by MLDE is plotted against the empirical fitness of the Sg5 and Sg8 on-target activity in the best-performing runs (ranked according to the NDCG and enrichment score). The top-5% hits in the prediction are highlighted in red, while the top-5% variants from the empirical data are outlined in black. Wild-type SpCas9 and other previously characterized variants (eSpCas9, Opti-SpCas9, and OptiHF-SpCas9) are labeled. **c** Performance of MLDE prediction on SpCas9's activity at NGN PAMs. NDCG score of MLDE runs using a combination of embeddings (Bepler/Georgiev) and models (p1 and p2) is plotted when 10, 20, 25, and 50% of training data was supplied ($n = 3$ independent MLDE runs for each data size). **d** Predicted versus empirical fitness of variants in the best-performing MLDE runs using 20% training data for each PAM (NGAT, NGCC, NGGG, and NGAT). The top 20% (12 variants) in the prediction are highlighted in red and the top 20% from the empirical data outlined in black. The parental variant, VRQR, and the variant SpG are labeled. Source data are provided in the Source Data file.

delivered by lentiviruses into reporter cell lines that stably expressed GFP and a sgRNA targeting the *GFP* gene sequence (Fig. 2a). Variants that generate indel-mediated disruption of the *GFP* sequence and its expression were enriched in the sorted bin with low GFP fluorescence (i.e., Bin A) as compared with the GFP-positive population (i.e., Bin B) (Fig. 2a, b). The mutated sequences on KKH-SaCas9 were retrieved using Illumina NovaSeq. The activities for the library of KKH-SaCas9 variants were plotted based on their relative enrichment in the sorted bin (Fig. 2c; Supplementary Data 6). Our experimental screening results revealed that variants harboring mutations at residues 888 and 889 of the WED domain and 988 and 989 of the PI domain were frequently detected among the top-5%-ranked variants with high on-target activities, while those carrying wild-type sequences at 887 of the WED domain and 985 and 986 of the PI domain more likely confer the enzyme with higher activity (Supplementary Fig. 8). From the library of variants, we identified that two of them (harboring N888Q and N888Q/A889S) exhibited higher activity than KKH-SaCas9 when paired with 2 out of 3 tested sgRNAs (i.e., sg1 and sg3); for the third sgRNA (i.e., sg2), the two variants showed comparable editing efficiency to KKH-SaCas9 (Fig. 2c, d). When using other 3 sgRNAs targeting the *GFP* sequence harboring nonpermissive PAMs for KKH-SaCas9 (i.e., NNNYRT), the library variants, including the N888Q and N888Q/A889S variants, showed minimal effect on disrupting GFP expression, indicating that the variants do not have relaxed constraint at those PAMs (Fig. 2e; Supplementary Data 6).

Comparison between our in silico prediction results and experimental screen data indicated that MLDE can be used to predict KKH-SaCas9's activity. Here, MLDE using the Georgiev embedding with the ensemble of random forest and SVM algorithm (parameter 2) showed the best performance (Fig. 3a, Supplementary Data 7). The enrichment in identifying top-performing variants reached about 6.7-fold for sg1, 9.2-fold for sg2, and 5.1-fold for sg3 with 20% input, and about 5.1-fold for sg1, 7.2-fold for sg2, and 4.1-fold for sg3 with 10% input performance (Fig. 3a, Supplementary Data 7). Although using the other parameters (i.e., Bepler.p1, Bepler.p2, and Georgiev.p1) also achieved high enrichment scores (Supplementary Figs. 9–11), our results indicated that these parameters gave more predicted variants with high enrichment score that were not enriched in the experimental datasets. Indeed, Georgiev.p2 parameter gave the best prediction performance across most datasets used in our ten in silico and experimental cross-validation work throughout our study. Our findings indicate the importance to select the best-performing embedding and modeling parameters for more consistent predictions in succeeding screens.

In addition, we noticed that for certain datasets that lack high-fitness variants in the training input could result in most variants from MLDE being predicted as depleted. Specifically, when training datasets only contained variants with poor activities

(<55% of the activity of the top experimentally validated variant), MLDE performance was hindered (Supplementary Figs. 9–11). Such condition was prominent in datasets of sg3 that 2 out of 3 training datasets failed to sample any high-fitness variants. Our results are in line with the recommendation that we ought to focus on surveying diverse sequence spaces believed to contain functional variants for MLDE[26]. Thus, the good performance of MLDE requires the presence of variants with higher fitness in the input training datasets.

Overall, with datasets that contained higher fitness variants for MLDE runs, we found that the three independent sets of activity measurements on KKH-SaCas9 variants using sgRNA sg1, sg2, or sg3 yielded consistent predictions with the experimental screen data, especially in MLDE predictions using the Georgiev embedding and modeling parameter 2 (Fig. 3b; Supplementary Fig. 8). This result is in line with our SpCas9 activity prediction showing that MLDE identifies top-performing variants readily. The top-5% hits predicted from the three sets included N888Q and N888Q/A889S variants identified in our experimental screen data (Fig. 3b). The high level of consistency, including the identification of the common top-performing variants, between the in silico and experimental screen data, confirms that the MLDE model can be used to predict KKH-SaCas9's variants with high activity.

To further verify the editing efficiencies of the identified variants with increased KKH-SaCas9's activity, individual validation assays were performed. The validation results were consistent with the screening data, from which we revealed that the N888Q variant exhibited increased editing activities over KKH-SaCas9 when paired with sg1 and sg3 sgRNAs (Supplementary Fig. 12). Together, our screen identifies residues located proximal to the PAM duplex that could be modified to increase KKH-SaCas9's on-target activity.

**Structure-guided engineering of activity-enhanced KKH-SaCas9-plus.** Based on the above-identified activity-enhanced variants, we explored using structure-guided engineering to further improve the editing activity of KKH-SaCas9. Protein-structure analyses indicated that N888 and A889 at the WED domain of SaCas9 are positioned close to its PI domain and the DNA backbone of the PAM duplex[18]. Our modeling revealed that while N888Q removes its contact with the DNA backbone of the PAM duplex, it could have increased its proximity to and added interaction with L989 at the PI domain (Fig. 4a)[31]. We speculated that this interaction may sandwich the PAM duplex more firmly to facilitate unwinding of the target DNA and trigger base pairing between the sgRNA and the DNA target, thereby explaining for the greater editing activity observed for the N888Q variant.

We tested whether switching N888 and A889 to other residues that could strengthen the interactions between WED and PI domains also enhances KKH-SaCas9's activity. We engineered

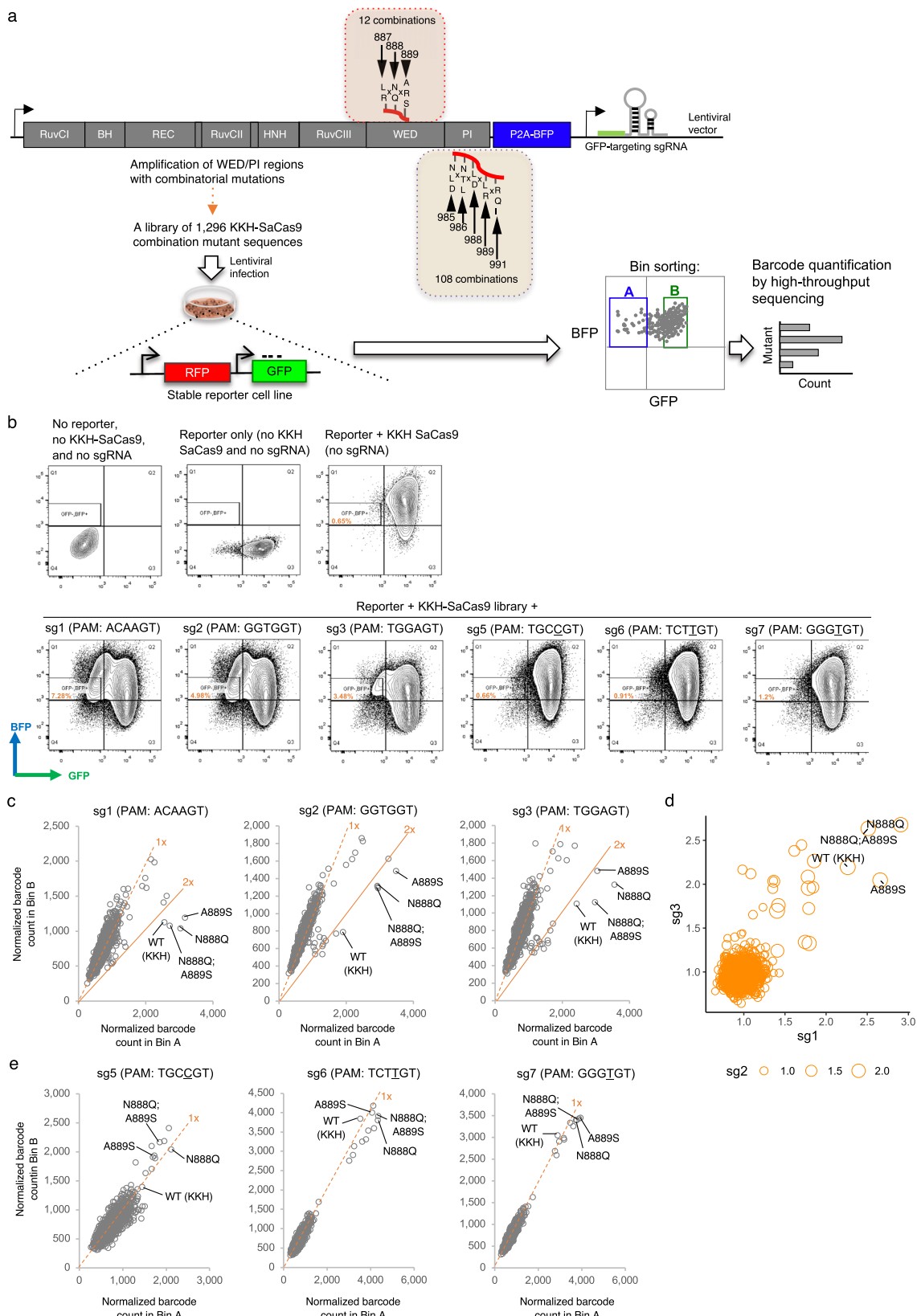

four more combined mutation variants (i.e., N888H/A889Q, N888S/A889Q, N888R/A889Q, and N888H/A889N) on those positions, which were selected based on predicted contact gains with the PI domain via N986, D987, L988, and/or L989 (Fig. 4a; Supplementary Fig. 13). Among the variants tested, the one harboring N888R/A889Q mutations (hereafter designated as KKH-SaCas9-plus) exhibited the greatest editing activity (i.e., 127% of KKH-SaCas9's activity, averaged from 3 sgRNAs targeting *GFP*) (Fig. 4b). Modeling showed that N888Q putatively only added contact with the PI domain via L989, while A889Q was predicted to interact with N986 and D987, as well as adding contacts with the DNA backbone of the PAM duplex (Fig. 4a).

**Fig. 2 Experimental screening of the activity of KKH-SaCas9 variants. a**, **b** Strategy for the profiling of the activities of KKH-SaCas9 variants in human cells is illustrated in (**a**). A library of 1,296 KKH-SaCas9 variants was assembled by PCR-based mutagenesis and was cloned in tandem with a gRNA-targeting GFP expressed from a U6 promoter. The library was delivered via lentiviruses to OVCAR8-ADR reporter cell lines in which the RFP and GFP genes are expressed from UBC and CMV promoters, respectively. Fluorescent protein expressions were analyzed by flow cytometry (results are shown in (**b**)). The activity of KKH-SaCas9 was measured using reporter systems in which the gRNA spacer sequence completely matched the GFP target site. Cells with an active KKH-SaCas9 variant were expected to lose GFP fluorescence. Cells were sorted into bins, each encompassing ~5% of the population based on GFP fluorescence, and their genomic DNA was extracted for quantification of the variant by Illumina NovaSeq. **c–e** Scatterplots comparing the barcode count of each KKH-SaCas9 variant between the bin-A (GFP-negative) and the bin- B (GFP-positive) populations. Each dot represents a KKH-SaCas9 variant, and wild-type (WT) KKH-SaCas9 is labeled. Solid reference lines denote 2-fold enrichment, and the dotted reference line corresponds to no change in barcode count in the bin-A as compared with the bin-B population. Three sgRNAs with permissive (sg1, sg2, and sg3) (**c**) and three sgRNAs with nonpermissive (sg5, sg6, and sg7) (**e**) PAMs for KKH-SaCas9 were used. Bubble plot summarizing the enrichment scores determined for each KKH-SaCas9 variant with the three sgRNAs with permissive PAMs is shown in (**d**).

We further confirmed that KKH-SaCas9-plus generated more edits when targeting endogenous genes (i.e., showed 109% of KKH-SaCas9's activity, averaged from sgRNAs targeting 7 loci) (Fig. 4c), while 3 out of the 7 loci showed 17–33% enhancement of the editing activity (Fig. 4d). We verified that the increase of editing activities observed was not due to the difference in the variants' protein expression (Fig. 4e). In addition, we observed that grafting N888R/A889Q mutations onto the KKH-SaCas9-derived cytosine base editor (BE4max) also increased its activity on editing 4 tested endogenous loci (by 11–93% at the most edited base within the target sites) (Supplementary Fig. 14), suggesting that the increased editing activity brought by the mutations is likely dictated at the DNA-binding level.

Modeling of KKH-SaCas9-plus showed that it contacts the PI domain via all four residues (i.e., N986, D987, L988, and/or L989) and has three contacts with the DNA backbone (Fig. 4a). Whereas, the other variants carrying N888H/A889Q and N888S/A889Q mutations could interact with the PI domain only via N986/987, but not L988/L989, with an equal number or more contacts with the DNA backbone (Supplementary Fig. 13). Hence, this illustrates that the creation of new interactions between the WED and PI domains at multiple locations within the PAM duplex region appears to be effective in augmenting KKH-SaCas9's activity, and accounts for the greater enhancement for KKH-SaCas9-plus.

## Discussion

There have been tremendous efforts in designing Cas9 proteins to boost gene editing efficiency at the same time purge undesired off-target editing. The two qualities involved maintaining a delicate balance of interacting and noninteracting amino-acid side chains of the Cas9 protein with the sgRNA–DNA complex. Dozens of variants possessing different mutation combinations have been reported thus far, each representing one of the many optimal solutions for the trade-off between Cas9 activity and precision. Considering that any of the amino-acid sites of SaCas9 in spatial proximity to the sgRNA–DNA complex are potential sites for optimization, which reaches over 40 sites[18], the number of combinatorial variants to screen through for optimization is too many (i.e., $2^{40} = 1.1 \times 10^{12}$) for wet-lab experiment even if each site is restricted to two (wild-type or mutated) amino-acid residues.

We have previously shown that with rational design, we could limit each site to 4–5 candidate residues and generate a targeted mutagenesis library to reduce screening efforts. Using that strategy, we successfully identified SpCas9 variants with both high activity and fidelity in a combinatorial screen of 952 variants[8]. Here, we explored how to facilitate such a rational-design-based screen with machine learning in the optimization of Cas9 proteins (Fig. 5). In particular, we sought to assess if ML can further downsize the experimental screen via the extrapolation of handful

of variants with experimentally determined fitness values. We found that ML-based in silico screen facilitates the search of more efficient Cas9 variants. In the best MLDE run on the SpCas9/Sg5 dataset, using as little as 20% of variants as input training data, we had a 51.5% chance of capturing the top variants according to experimental data, compared with the chance in the null model. The synthetic data, generated by MLDE trained with experimental data of only 130 variants, allowed the identification of 17 top-performing (5%) variants, achieving a 3.8-fold increase in resource efficiency to 0.131 (i.e., 17/130) compared with 0.035 of the empirical approach (i.e., 33/952, 33 top-5% variants were taken from the 650 available datapoints for comparison) where the full library (with 952 variants) was screened (Supplementary Data 8).

Shortlisting a few candidate residues on selected amino-acid sites via structure-guided rational design of SpCas9 has already enhanced our chances of finding better variants in our previously published combinatorial mutant library. Likewise, MLDE recommended that we ought to focus on surveying diverse sequence spaces believed to contain functional variants (ftMLDE mode[26]). Thus, we tested how the MLDE performance would differ when training data was selected randomly or selected to maximize sequence diversity among variants. We found that the training datasets with divergent variants conferred negligible benefits on MLDE performance.

In an independent Cas9 optimization task, we further demonstrated that MLDE exhibited surpassing performance in the prediction of KKH-SaCas9 variants' activities on three sgRNAs and showed success in identifying useful variants in the KKH-SaCas9 screen subsequently. In our best-performing ML runs on the KKH-SaCas9 datasets, using 20% of variants (260 out of 1,296 variants) as input training data resulted in a 40.0, 49.2, and 44.6% chance of capturing the top-5% variants for sg1, sg2, and sg3, respectively, according to experimental data (Supplementary Fig. 8). Screening only 260 out of 1,296 variants for generating the training dataset led to the identification of 26–32 top-performing variants using MLDE. The resource efficiency of identifying the top-performing (5%) variants was increased by 2.0- to 2.5-fold to 0.100 (i.e., 26/260), 0.123 (i.e., 32/260), and 0.112 (i.e., 29/260) for sg1, sg2, and sg3 respectively, compared with 0.050 (i.e., 65/1,296) in the null model. We combined structure-guided design, targeted mutagenesis library screen, and ML in this study to identify activity-enhanced KKH-SaCas9 variants, vastly shortening the path to identify these top variants.

The best-performing variant, KKH-SaCas9-plus, reported in this work harbors N888R/A889Q mutations that improve its editing activity. Our molecular modeling provides structural insights that these mutations may have strengthened the interactions between KKH-SaCas9's WED and PI domains located near the PAM duplex to anchor the target DNA in the SaCas9–sgRNA-target DNA complex. While N888R/A889Q

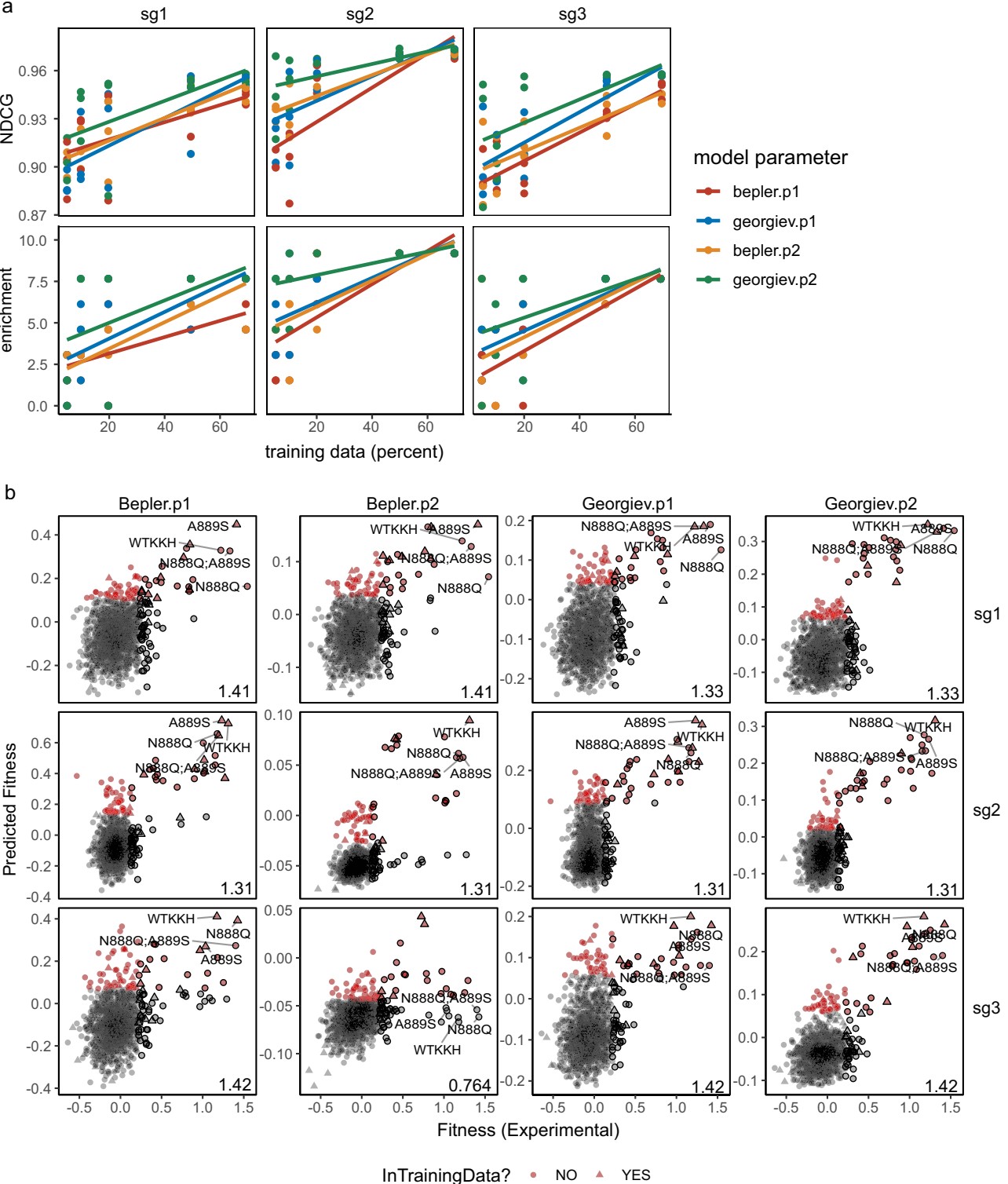

**Fig. 3 MLDE predictions on KKH-saCas9's on-target activity with three sgRNAs. a** Enrichment and NDCG of MLDE runs using a combination or embeddings (Bepler/Georgiev) and models (p1 and p2) are plotted against the size of training data. The best-fit line summarizes 3 replicates for each embedding and model parameter combination of MLDE runs using 5, 10, 20, 50, and 70% of training data. **b** Predicted versus empirical fitness of variants in the best-performing MLDE runs using 20% input training data, given the different combinations of embeddings and model parameters. The predicted fitness by MLDE is plotted against the empirical fitness of the on-target activity of KKH-SaCas9 with three sgRNAs (sg1, sg2, and sg3) in the best-performing runs (ranked according to the NDCG and enrichment score). The values of maximum fitness in the training data are indicated at the bottom-right corner of each panel. The top-5% hits in the prediction are highlighted in red, while the top-5% variants from the empirical data are outlined in black. Wild-type KKH-SaCas9 (WT-KKH) and top-performing variants N888Q, N888Q/A889S, and A889S are labeled. The source data are provided as a Source Data file.

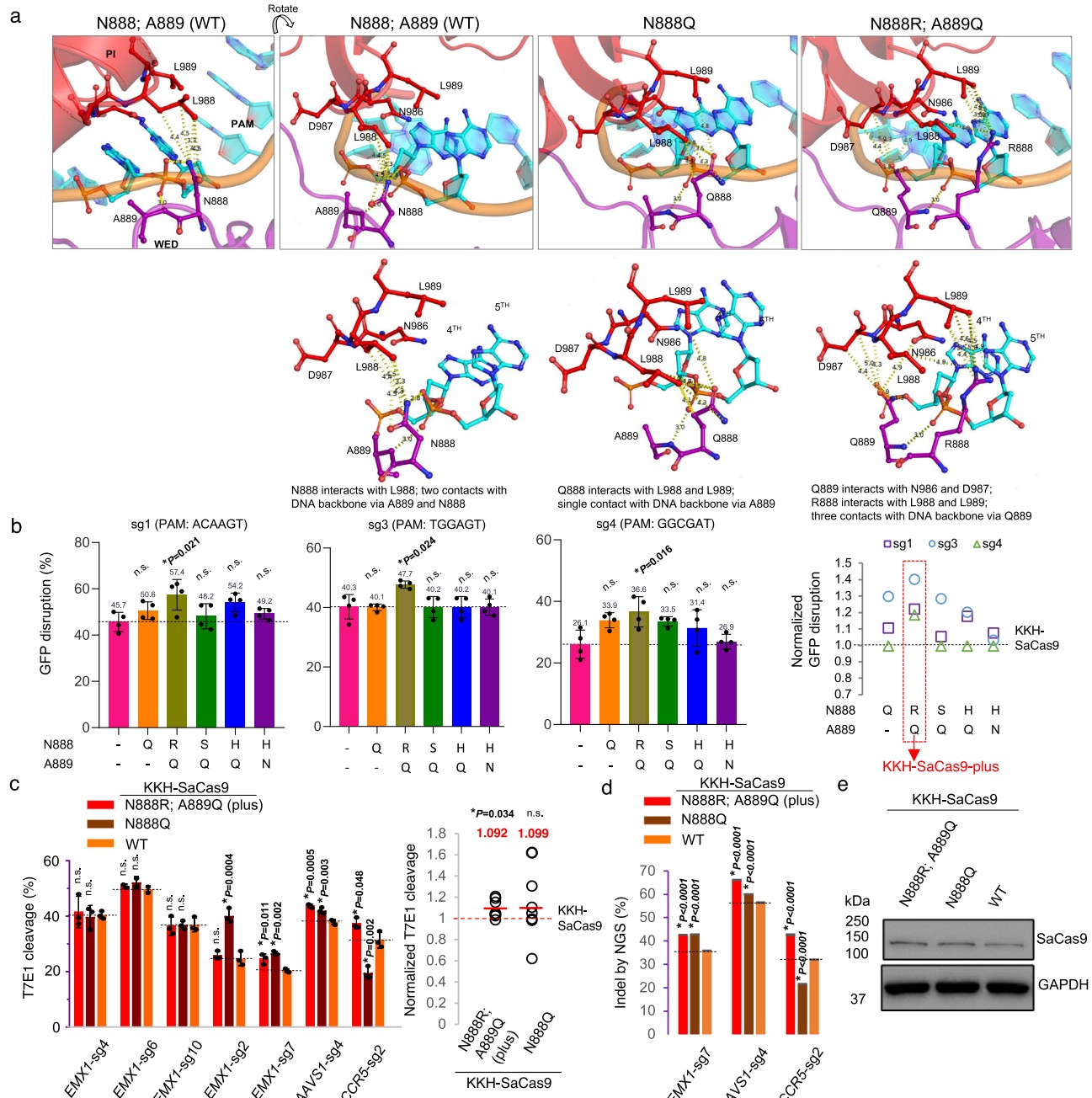

**Fig. 4 Structure-guided engineering improves the editing efficiency of activity-enhanced KKH-SaCas9 variants. a** Molecular modeling of N888Q and N888R/A889Q mutations on WED domain of SaCas9 depicts their increased interactions with residues on its PI domain and the DNA backbone. **b** KKH-SaCas9 variants carrying mutations on residues 888 and/or 889 were individually constructed and characterized using GFP disruption assays with three independent sgRNAs at 5 days post infection. The mean editing efficiency +/− SD (error bar) of the KKH-SaCas9 variants ($n = 4$ biologically independent samples) was measured as the percentage of cells with depleted GFP fluorescence using flow cytometry. Statistical significance was analyzed by one-way ANOVA with Tukey's test. The $P$-values of 0.016–0.024 indicate the comparisons with the wild-type variant's activity. **c**, **d** Assessment of KKH-SaCas9 variants' on-target editing with sgRNAs targeting endogenous loci. The percentage of sites with indels was measured using a T7 endonuclease-I (T7E1) assay in (**c**) and deep-sequencing assay in (**d**). Mean and standard deviation are shown for the loci tested. Each locus was measured in $n = 3$ biological independent samples. Statistical significance was analyzed by one-way ANOVA with Tukey's test. The $P$-values of <0.0001–0.048 indicate the comparisons with the wild-type variant's activity. The ratios of the on-target activity of KKH-SaCas9 variants with N888Q and N888R/A889Q mutations to the activity of KKH-SaCas9 were determined, and mean for the normalized activity is shown and highlighted by a red line ($n = 7$, one-sample t-test). n.s. indicates not significant. **e** Western blot analysis on protein expression of the KKH-SaCas9 variants. This experiment was performed once. The source data for figures (**b**–**e**) are provided as a Source Data file.

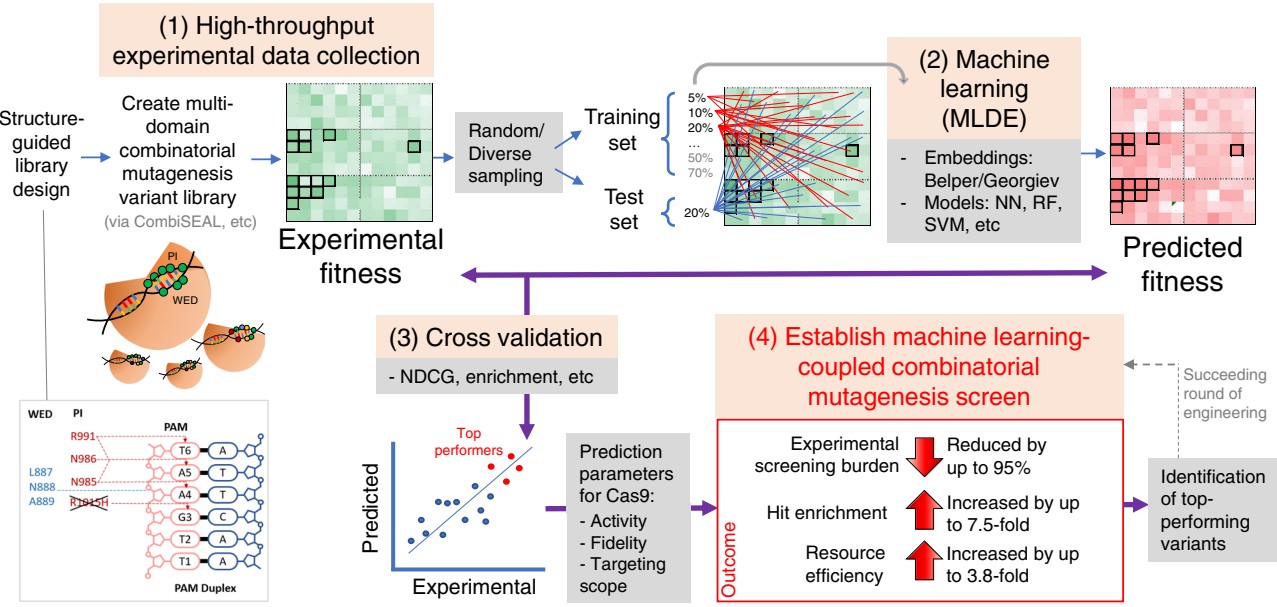

**Fig. 5 Machine-learning-integrated multidomain combinatorial mutagenesis screen for resource-efficient protein engineering.** An overview of our study procedures and outcomes. We started with structure-guided design to select sites and residues for mutagenesis and built multidomain combinatorial variant libraries. We then run MLDE and tested embedding and model parameters to generate in silico predictions. We cross-validated experimental and MLDE predictions, which established parameters for accurate prediction of Cas9's activity, fidelity, and targeting scope. The machine-learning-coupled multidomain combinatorial mutagenesis screening approach facilitates the identification of top-performing variants with much reduced experimental screening burden, increased hit enrichment, and enhanced resource efficiency.

increases the on-target activity of KKH-SaCas9, it may increase off-target editing. Our GUIDE-seq results indicated that KKH-SaCas9-plus showed a comparable on-to-off-target editing ratio to wild type, albeit that there were alternative off-target sites identified (Supplementary Fig. 15a, b). KKH-SaCas9-plus showed more off-target edits at some of the target sequences with single-base mismatches (Supplementary Fig. 16). We also tested whether the addition of N888R/A889Q could improve the activity of the more accurate variant of KKH-SaCas9 (i.e., SAV2)[16]. We found that N888R/A889Q also enhanced the on-target activity of SAV2 (i.e., showed 121% of SAV2's activity, averaged from sgRNAs targeting 8 loci), while 5 out of the 8 loci showed 9–48% enhancement of the editing activity (Supplementary Fig. 15c, d). This combined mutant (KKH-SaCas9-SAV2-plus) generated comparably few genome-wide off-target edits (Supplementary Fig. 15a, b), while we observed increased edits at some of the tested off-target sites with single mismatches (Supplementary Fig. 16). These results indicate the feasibility to combine activity- and specificity-enhancing mutations for further optimizing the KKH-SaCas9's performance. This result also affirms that the abilities of KKH-SaCas9 to bind the DNA and distinguish base mismatches between sgRNA and the DNA target probably act through distinct mechanisms, and thus its activity and specificity could be engineered independently. Furthermore, our data indicated that the increased editing activity brought by the mutations is likely dictated at the DNA-binding level, it is plausible that N888R/A889Q may also be compatible with other dSaCas9-derived genome-perturbation tools, including gene activators[32,33], base editor[34,35], and prime editor[36] via increasing their abilities to bind the DNA and thus their activities. The N888R/A889Q mutations on the WED domain represent a useful building block for further engineering of various genome-perturbation tools to achieve both high activity and specificity. Future comprehensive analyses using a large panel of sgRNAs paired with their target sequences will help define the target-sequence dependencies and sgRNA design rules for the KKH-SaCas9-plus variants.

To discover the activity-enhanced KKH-SaCas9 variants, we initiated in this work a smaller pool of (about a thousand) variants to be experimented with based on structure-guided design. Except for the datasets that we generated previously on SpCas9, other Cas9-engineering studies relied on methods, including random and site-directed mutagenesis to select limited (often only tens of) clonal isolates for characterization, and there is a lack of large-scale experimental screening datasets available for ML. To aid our evaluation of the ML methods for engineering KKH-SaCas9, here we generated large-scale datasets for this. This initial work is important because we unexpectedly noticed that the selection of suitable sgRNAs (e.g., Sg5 for SpCas9 and sg1, sg2, and s3 for KKH-SaCas9) could allow MLDE to generate more reliable predictions in subsequent screens (see Supplementary Text). We tested and validated the MLDE-based workflow based on our experimental screening data and defined the required number and diversity of the input combinations for in silico predictions. Our results set the groundwork to prepare for succeeding screens of many more combinatorial mutations via the creation of a directed library at a manageable experimental scale. While MLDE was successful in predicting Cas9 variants with high activity, it did not precisely inform which variant among the predicted top-performing ones ranks best. A further step is needed to experimentally characterize the shortlisted top-performing candidates. Continual efforts in advancing ML methods for protein-structure modeling, including incorporating structural descriptors[37] in addition to the learnt representation, to improve the prediction on variants' activities should further enhance future in silico screens. Meanwhile, we only investigated mutation combinations from selected amino-acid residues by rational design and did not explore the performance of MLDE on a virtual fully saturated mutagenesis screen. Creating a more comprehensive screening strategy involves designing a library enriched with diverse but not dysfunctional variants, this remains a challenge to be addressed. One could examine possible structural changes of the designed variants predicted using other in

silico tools such as DynaMut[31], Rosetta[38,39], and Pymol to further filter for candidate mutations. For example, experimental screening of a computationally designed library of ubiquitin variants was shown to be more successful in identifying variants with strong protein-binding ability[40].

Another direction we would be eager to pursue is to increase the number of amino-acid sites we can survey. It would be particularly useful for protein repurposing to use another substrate, where the wild type has essentially no activity. An example would be further pushing for a SaCas9 with much relaxed PAM constraint, which may involve engineering multiple sites beyond the PI and WED domains. The number of targeted mutagenesis sites to incorporate is still a confounding factor in combinatorial library construction. For example, commercial oligo synthesis of a 100-bp DNA fragment at most accommodates 10 sites of NNN/NNK degenerate codons or trinucleotide pool. MLDE has the potential to transcend such physical limitations by building a combined in silico screen supplied with empirical data from multiple smaller focused libraries. One possibility is to perform multiple focused screens with MLDE-converging sites with modest overlaps that each library has mutagenesis of 5 amino-acid residues per site, up to 6 sites with 1–2 sites in common to another library (Supplementary Fig. 17a). Then, we can use MLDE to combine all these experimental data in silico to predict the optimal variants. Another possibility is performing iterative rounds of targeted mutagenesis (Supplementary Fig. 17b). The best variants we found at the end of each round seed the mutagenesis library of the next round with a new set of amino-acid sites subjected to mutagenesis. In both screening schemes, MLDE and other ML-based methods will play an important role in the search for high-performance variants and will serve as an invaluable tool in the toolkit of protein engineering. Complementary methods, including polymerase- chain reaction-based mutagenesis and CombiSEAL[8,41] that allow assembly of combinatorial mutations scattered over the entire protein, will facilitate desirable targeted mutagenesis libraries to be built and experimented.

## Methods

**Generation of data input for MLDE**. We used the previously published SpCas9 data[8] surveying the on-target activity of Sg5 (650 empirical datapoints) and Sg8 (729 empirical datapoints) that target a red fluorescent protein (RFP) sequence as the input data for the MLDE package. Specifically, we used the dataset of on- and off-target activities measured from the same CombiSEAL library of SpCas9 variants engineered at 8 amino-acid positions (R661, Q695, K848, E923, T924, Q926, K1003, and R1060) that interact with the sgRNA–DNA complex. The on-target activity was measured in screens where the sgRNA and target site have perfectly matched protospacer sequences, while off-target activity was measured where the sgRNA is targeting a site that bears an artificially introduced synonymous mutation. We removed the extreme outlier by setting the minimal enrichment score (E-score) as −2 for both Sg5 and Sg8 before the min–max normalized to the scaled fitness score ranging between 0 and 1. First, we isolated 20% of the entire library (190 SpCas9 variants) as test data; among these selected variants, 122 of Sg5 and 136 of Sg8 have empirical measurements. We then generated input training datasets that do not overlap with the test data. The training datasets consist of 5%, 10%, 20%, 50%, and 70% of randomly drawn empirical measurements to test the minimal input for effective selection of top variants from MLDE prediction, corresponding to datasets of 33, 65, 130, 325, and 445 empirically measured Sg5 on-target activity and 37, 73, 146, 365, and 510 Sg8 on-target activity measurements. We generated three replicates for each size, subjected to either randomized or diverse selection schemes for variants. To generate the randomized dataset, we used the sample_n() function from dplyr in R to randomly select the predefined number of E-scores. Taking the above-mentioned 20% of the entire library as non-overlapping test data allows the 70% randomly selected data not being the same for all three replicates, while maximizing the variant numbers for evaluation. Using the same method, we prepared the datasets of Sg5 off-target activities for MLDE. First, we withheld 190 variants as the test set for the off-target activities. Then we randomly sampled the remaining variants that consist of 5%, 10%, 20%, 50%, and 70% of the library to generate 3 replicates of training datasets for each size, corresponding to 41, 83, 165, 414, and 579 empirical measurements of Sg5 off-target activity. The off-target activity was derived from min–max normalized E-score after setting a lower bound of −2.5.

We sought to increase the sequence diversity of the input training data by restricting the presence of variants, whose share only 1 or 2 mismatches (mutations apart) with each other, to the minimum, given the data size. To do so, we kept randomly sampling variants to size N with available E-scores until each variant sharing no more than p1-mismatch neighbors and q2-mismatch neighbors in the subgroup N. The thresholds p and q for each dataset are listed in Supplementary Data 1. As N increases in size, it is more difficult to remove the 1- and 2-mismatch neighbors in the input dataset and the overall sequence diversity of the diverse dataset becomes similar to that of the randomly subsampled input datasets. Consequently, we generated diverse datasets of four different sizes for Sg5 and Sg8 on-target activity that correspond to approximately 5, 10, 20, and 50% of the empirical data. Because the 50% training datasets showed the same level of diversity as randomly selected data, we only run MLDE on 5, 10, and 20% datasets. The resultant diversity of the dataset, summarized as the total number of pairwise sequences with N mismatches, is listed in Supplementary Data 1.

We also used the dataset with a total of 58 SpCas9 variants bearing rational substitutions at five positions located in the PI domain that had their activities on noncanonical NGN PAMs assessed by HT-PAMDA[30]. The on-target activity of the variant against 4 sgRNAs representing NGAT, NGCC, NGGG, and NGTA PAMs was min–max normalized in the training data. To avoid having too few variants in the test set given the small dataset size, we withheld 29 variants (50% of the library) as test data and performed MLDE with combinations of Bepler and Georgiev and modeling parameters p1 and p2 to predict on-target activity predictions using 10, 20, 25, and 50% input (empirical data of 29 and 15 variants).

The in-house SaCas9 dataset consists of 1,296 variants that were constructed and tested in this study. Substitutions on 8 amino-acid positions (887, 888, 889, 985, 986, 988, 989, 991) that are widely scattered over the WED and PI domains were rationally chosen based on protein-structure analyses (see Supplementary Data 3 for details). The SaCas9 variants' on-target activities against sgRNA 1, 2, and 3 were measured as the E-score derived from the high-throughput fluorescent protein disruption assay. We again withheld 20% of the empirical data (260 variants) as the test set. From the remaining variants, we generated 3 replicates of randomly selected datasets that consisted of 65, 130, 260, 648, and 907 variants that correspond to 5, 10, 20, 50, and 70% of the full library as training data for MLDE. We run MLDE using the training data of different sizes and evaluate the MLDE performance using the test-set variants.

We run MLDE according to the default parameters. Briefly, we applied the Bepler and Georgiev embedding of the full-length amino-acid sequences of SpCas9 (UniProtKB—Q99ZW2 (CAS9_STRP1)) [https://www.uniprot.org/uniprot/Q99ZW2] and SaCas9 (UniProtKB—J7RUA5 (CAS9_STAAU)) [https://www.uniprot.org/uniprot/J7RUA5] substituted with the designated variant's amino-acid residue combination. We modified the MLDE GenerateEncodings.py so that it processed a customized input fasta file containing the protein sequences of all the variants designed in the SpCas9 as well as the SaCas9 dataset rather than generating the full set of saturated mutagenesis variants. We run the MLDE ExecuteMlde.py with default parameters on the Bepler and the Georgiev embeddings and with two different sets of parameters. We assigned them as parameters 1 and 2: parameter 1 used the neural network models "NOHidden", "OneHidden", "TwoHidden", "OneConv", and "TwoConv" available in MLDE, each with 20 rounds of hyperparameter optimization, and parameter 2 used less complex models "Linear-Tweedie", "RandomForestRegressor", "LinearSVR", and "ElasticNet", each with 50 rounds of hyperparameter optimization.

We evaluated the performance of the ML algorithm parameters and embeddings with precision, specificity, and sensitivity, using the withheld test-set variants. More specifically, we assign variants with at least 70% of the wild-type activity as positives and the rest as negatives. Thus, true positives are variants with at least 70% activity of the wild type when empirically tested with the sgRNA. Otherwise, they are true negatives. For each MLDE result, we also labeled the positives and negatives using the 70% wild-type activity threshold. We then counted the number of true positives (TP), true negatives (TN), false positives (FP), and false negatives (FN) for each result and derived the performance metrics according to the formula stated below:

$$\text{specificity} = \frac{\text{TN}}{\text{TN} + \text{FP}} \qquad (1)$$

$$\text{sensitivity} = \frac{\text{TP}}{\text{TP} + \text{FN}} \qquad (2)$$

$$\text{Precision} = \frac{\text{TP}}{\text{TP} + \text{FP}} \qquad (3)$$

We also applied another performance metric, enrichment, proposed by Sarfati et al.[42]. The enrichment reports the ratio of identifying true top 5% of hits when using the ML prediction to random selection (the null background)

$$\text{Enrichment} = I_5^{\text{prediction}} / I_5^{\text{random}} = \frac{400 \cdot I_5^{\text{prediction}}}{N}, \qquad (4)$$

where $N$ is the total size of the test set, here N is the number of all the variants in the prediction. Enrichment provides us with an estimate of identifying high-fitness variants when we select the top-5% variants by predicted fitness for downstream

experimental validation. When the larger fraction of highest-fitness variants is captured in the top-5% prediction, enrichment increases from 1.

Finally, NDCG is also calculated to evaluate how well the model identifies top-ranking variants

$$\text{NDCG} = \left( \sum_{i=1}^{N} \frac{f_{\text{rel } i}}{\log_2(\text{rel } i + 1)} \right) \bigg/ \left( \sum_{i=1}^{N} \frac{f_i}{\log_2(i + 1)} \right), \quad (5)$$

where $f$ is the true fitness value of the variant, $i$ is the true rank (from highest to lowest fitness), and rel $i$ is the predicted rank from the model. NDCG compares the predicted ranking to the actual ranking, aligns with the goal of MLDE to identify high-fitness variants as top-ranking variants[24,25]. If the predicted ranking and the actual ranking is identical, NDCG reaches its maximum value of 1. Models that misidentify low-fitness variants as top-ranking ones would result in low NDCG.

The input data handling, statistical analyses, and graph plottings are carried out in R v4.1.2 using packages ggplot2 v3.3.5, tidyverse 1.3.1, readxl, Cairo, and stringdist.

**Plasmid construction**. The plasmids generated from this study (Supplementary Data 9) were done with standard molecular cloning techniques such as PCR, restriction-enzyme digestion, ligation, one-pot ligation, and Gibson assembly. Customized oligonucleotides were ordered through Genewiz. Vectors were transformed into E. coli strain DH5α-competent cells and selected with ampicillin (100 mg/ml, USB) or carbenicillin (50 mg/ml, Teknova). DNA was extracted and purified by Plasmid Mini (Takara and Tiangen) or Midi preparation (QIAGEN) kits. Sequences of the vectors were verified with Sanger sequencing.

Storage vectors AWp28 (Addgene #73850) and AWp112 were used to assemble the sgRNA chosen to target a specific gene. The sgRNA sequences used are listed in Supplementary Data 10. Oligonucleotide pairs of the sgRNA target sequences with BbsI sticky ends were synthesized, annealed, and cloned into the BbsI-digested storage vector using T4 DNA ligase (New England Biolabs). To prepare the lentiviral vector for SaCas9 variant expression, AWp124 vector was modified via Gibson assembly to remove all existing Esp3I enzyme sites. Esp3I sites were then reintroduced flanking the PI and WED regions to incorporate the intended mutations, giving the DTp2 vector. To insert the sgRNA expression cassette, they were amplified from the storage vector with flanking BamHI and EcoRI (Thermo Fisher Scientific) sites to and ligated with the digested lentiviral vector DTp2. To generate the PI and WED mutations, oligonucleotides with the WED-domain mutations were pooled in a 1:1 ratio as the forward primer, and the same was applied with the PI domain for the reverse primer. PCR amplifications were done using these pooled forward and reverse primers with the original KKH-SaCas9 template to create the pooled mutations. Using a one-pot ligation method, the pooled mutations were inserted into the Esp3I sites of DTp2. The EFS promoter drives the SaCas9 expression, together with a fluorescent protein expression from the downstream T2A-BFP. To create SaCas9-KKH-SAV2-plus (DTp47A), we incorporated the Esp3I sites similarly done with DTp2 into SaCas9-KKH-SAV2 (DTp52) via Gibson assembly, and then with one-pot ligation inserted the 'plus' mutations that are the N888R/A889Q. All plasmids created in this study are available from the authors.

**Cell culture and transduction**. HEK293T cells obtained from American Type Culture Collection (ATCC), and MHCC97L-Luc cells gifted by S. Ma (School of Biomedical Sciences, The University of Hong Kong), were maintained in Dulbecco's Modified Eagle Medium (DMEM) supplemented with 1× antibiotic–antimycotic and 10% FBS (Thermo Fisher Scientific). OVCAR8-ADR cells gifted by T. Ochiya (Japanese National Cancer Center Research Institute, Japan), were maintained in RPMI 1640 medium supplemented with 10% FBS (Gibco). The HEK293T cells were used for lentiviral production for KKH-SaCas9 variant expression and for generating stable cell lines. OVCAR8-ADR cells were transduced with a pAWp9 vector (Addgene #73851) expressing RFP and GFP gene, driven by the hUbCp and CMV promoters, respectively, for the initial screening of KKH-SaCas9 pooled variants and for further validation. OVCAR8-ADR cells were also transduced with lentiviruses encoding RFP and GFP genes expressed from UBC and CMV promoters, respectively, and a tandem U6 promoter-driven expression cassette of sgRNA targeting the GFP site. For the initial screening, the KKH-SaCas9 variants were expressed with sgRNA targeting GFP using EFS and U6 promoters, respectively, followed by a T2A-BFP to determine KKH-SaCas9 expression. The cells were sorted with a Becton Dickinson BD Influx cell sorter. With the mutational screening, the KKH-SaCas9-selected variants were transduced into the stable OVCAR8-ADR cell lines harboring the GFP, RFP genes, and sgRNA. The MHCC97L-Luc cell lines were transduced to create the stable expression of the selected KKH-SaCas9 variants for the T7E1 and GUIDE-seq experiments. The cells were regularly tested and showed negative for mycoplasma contamination. Lentivirus production and transduction were carried out as previously described[8].

**Fluorescent protein disruption assay**. Fluorescent protein disruption assays were conducted to determine DNA cleavage and indel-mediated disruption at the target site of the fluorescent protein, GFP, by the KKH-SaCas9 variants with the gRNA expressions, resulting in loss of cell fluorescence. The stable cell lines integrated with the GFP and RFP reporter gene, expressing the SaCas9 variants and sgRNA, were washed, resuspended with 1× PBS supplemented with 2% heat-inactivated FBS, and analyzed with Becton Dickinson LSR Fortessa Analyzer or ACEA NovoCyte Quanteon. Cells were gated on forward and side scatter, and at least $1 \times 10^4$ cells were recorded per sample for each dataset. FlowJo v10.7 was used to analyze data generated from flow cytometry experiments.

**Immunoblot analysis**. Immunoblots were carried out as previously described[8]. Anti-SaCas9 (1:1000, Cell Signaling #85687) and anti-GAPDH (1:5000, Cell Signaling #2118) primary antibodies were used, followed by HRP-linked anti-mouse IgG (1:10,000, Cell Signaling #7076) and HRP-linked anti-rabbit IgG (1:20,000, Cell Signaling #7074) secondary antibodies. The unprocessed scan of the immunoblots is available in the Source Data file.

**T7 endonuclease-I assay**. T7 endonuclease-I assay was performed as previously described to quantify the Cas9-induced mutagenesis in endogenous loci[8]. The targeted loci were amplified from 15 to 30 ng of genomic DNA extracted using dNeasy Blood and Tissue Kit (QIAGEN) using the primers as listed in Supplementary Data 11. Quantification was based on relative band intensities measured using ImageJ. Editing efficiency was estimated by the formula

$$100 \times (1 - (1 - (b + c)/(a + b + c))^{1/2}), \quad (6)$$

as previously described[43], where $a$ is the integrated intensity of the uncleaved PCR product, and $b$ and $c$ are the integrated intensities of each cleavage product.

**GUIDE-seq**. GUIDE-seq was performed as previously described[8]. Approximately 1.6 million MHCC97L cells stably expressing the KKH-SaCas9 variants were transduced with sgRNAs. After 72 h, electroporation was conducted according to the manufacturer's protocol using 1100 pmol freshly annealed end-protected dsODN with 100 μl Neon tips (Thermo Fisher Scientific). The dsODN oligonucleotides used were 5′-P-G*T*TTAATTGAGTTGTCATATGTTAATAACGG T*A*T-3′ and 5′-P-A*T*ACCGTTATTAACATATGACAACTCAATTAA*A*C-3′, where P represents 5′ phosphorylation and asterisks indicate a phosphorothioate linkage. Electroporation voltage, width, and the number of pulses were 1100 V, 20 ms, and 3 pulses, respectively. Cells were harvested at day 7 post transduction of the sgRNA. Genomic DNA was extracted using dNeasy Blood and Tissue Kit (QIAGEN) according to the manufacturer's protocol. The gDNA collected for the SaCas9 variant and the sgRNA were sequenced on Illumina NextSeq System and analyzed with GUIDE-seq software[44].

**Deep sequencing**. Deep sequencing was carried out as previously described[45]. The same gDNA samples used for the T7E1 assays were amplified for the region of edit and sent for deep sequencing. About ~0.6 million reads per sample on average were used to evaluate the genomic diversity of the >10,000 cells. HEK293T cells were infected with sgRNAs and then transfected with KKH-SaCas9-derived BE4max editor, together with a fluorescent protein expression from the downstream T2A-BFP. The cells harboring both base editor and sgRNA were sorted with a cell sorter based on fluorescence. Amplicons harboring the targeted endogenous loci were generated by PCR. About ~0.2 million reads per sample on average were used to evaluate the genomic diversity of the >10,000 cells. Crispresso2[46] with default setting was used to quantify indels and base editing outcomes from the deep-sequencing data.

**Molecular modeling**. Molecular dynamic simulations were conducted on the variants using DynaMut[31]. The variant mutations were singly inputted into the webserver, and the structural outputs were then aligned with the crystal structure of SaCas9 (PDB: 5CZZ) [https://www.rcsb.org/structure/5CZZ] on PyMol. The predicted rotamer of the mutations as indicated by DynaMut was then used to replace the amino-acid positions on the SaCas9 crystal structure. The predicted interactions determined by DynaMut and Pymol were then indicated on the crystal structure to provide a putative representation of the SaCas9 variants.

**Reporting summary**. Further information on research design is available in the Nature Research Reporting Summary linked to this article.

## Data availability

The deep-sequencing data generated in this study have been deposited in the NCBI SRA database under accession code PRJNA817034. The GUIDE-seq data generated in this study have been deposited in the European Nucleotide Archive (ENA) database under accession code PRJEB51773. For analysis on previously published datasets, Choi et al.'s datasets[8] (including SpCas9 variants' activities on Sg5 on-target, Sg8 on-target, and Sg5 off-target sites) were retrieved from Supplementary Table 2 at https://www.nature.com/articles/s41592-019-0473-0#Sec24, and Walton et al.'s datasets[30] (including SpCas9 variants' activities on four noncanonical NGN PAMs) were retrieved from Supplementary Table S4 at https://www.science.org/doi/10.1126/science.aba8853. The previously released structural data used in this study: SpCas9 (UniProtKB—Q99ZW2 (CAS9_STRP1)) [https://www.uniprot.org/uniprot/Q99ZW2], SaCas9 (UniProtKB—J7RUA5 (CAS9_STAAU)) [https://www.uniprot.org/uniprot/J7RUA5], and SaCas9 (PDB: 5CZZ) [https://www.rcsb.org/structure/5CZZ]. Source data are provided with this paper.

## Code availability

The customized MLDE code and instructions are stored in github [https://github.com/AWHKU/RunMLDE_SpCas9].

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

## Acknowledgements

We thank members of the Wong lab for helpful discussions. We thank the Centre for PanorOmic Sciences at the LKS Faculty of Medicine, The University of Hong Kong, for providing and maintaining the equipment needed for flow cytometry analysis and cell sorting. We also thank the Information Technology Services at The University of Hong Kong for maintaining and providing support on utilizing the High Performance Computing System to process our data and analysis. This work was supported by the National Natural Science Foundation of China Excellent Young Scientists Fund (32022089), the Hong Kong Research Grants Council [17104619], and the Centre for Oncology and Immunology Limited under the Health@InnoHK Initiative funded by the Innovation and Technology Commission, The Government of Hong Kong SAR, China (to A.S.L.W.). A.S.L.W. is a Ming Wai Lau Centre for Reparative Medicine (MWLC) Associate Member, and this project was in part supported by the Ming Wai Lau Centre for Reparative Medicine Associate Member Programme. This work was also in part supported by AIR@InnoHK administered by Innovation and Technology Commission, The Government of Hong Kong SAR, China (to J.W.K.H.).

## Author contributions

D.G.L.T., H.Y.C, and A.S.L.W. conceived the work, designed the experiments, interpreted, and analyzed the data. D.G.L.T. performed molecular modeling and most of the experiments, and H.Y.C. performed the machine learning and data analysis. J.H.C.F., B.K.C.C., P.Z., C.C.S.K., and G.C.G.C. provided support for molecular biology experiments. Y.M.C., S.Y.L.M., and Z.Z. provided support for GUIDE-seq experiments. J.W.K.H. provided advice for machine learning. D.G.L.T., H.Y.C, and A.S.L.W. wrote the paper.

## Competing interests

A patent application (63/268,745) has been filed by the University of Hong Kong on the SaCas9 variants described here (inventors: A.S.L.W. and D.G.L.T.). All other authors declare no competing interests.

## Additional information

**Peer-review information** *Nature Communications* thanks Jiazhi Hu and the other, anonymous, reviewer(s) for their contribution to the peer review of this work. Peer reviewer reports are available.

