## [Peer Review File · Nature Communications]

Reviewers' Comments:

Reviewer #1:

Remarks to the Author:

The study by Dawn et al., reported the generation of activity-enhanced SaCas9 based on machine learning. The study used a machine learning package (MLDE) developed for predicting combinational protein fitness landscape to predict which combination of mutations can enhance Cas9 activity. The study first tested the MLDE with different sub-sets from a combinational mutation dataset (650 variants) of SpCas9 generated by the group in a previous study. The study found that with 10-20% of the variants as training dataset, the MLDE can predict functional variants with over 5-fold enrichment. The study then selected eight residues (three at the WED domain and five at the PI domains) at the KKH-SaCas9 protein. A total of 1296 combinational variants were designed. Combines MLDE prediction with 300 randomly picked data and experimental data generated by lentiviral library-based screening, the study showed that MLDE predicted SaCas9 variants with high activity is well correlated with those measured experimentally. The combined screening identified two variants exhibited marginal increased activity (see comments below), whereas the significance of increase was not documented yet. The study then applied structured-guided strategy to further improve the interaction of the two residues (N888 and A889) with residue in the PI domain, reasoning that increasing the interaction between WED domain the PI domain will increase the interaction between SaCas9 and the PAM. Four combined variants were tested, and one variant (N888R/A889Q) exhibited enhanced editing activity. Depending on the gRNAs tested, the activity was increased range from 11% to 30%, although whether these increases were significant or not remained to be clarified by the authors. And lastly, the study introduced the same mutations to an unpublished high-fidelity SaCas9 variant from the group. Similar degree of increased activity was observed. The study also applied GUIDE-seq to evaluate the off-target effects. Based on the results, the author concluded that the modification increases activity whilst does not compensate specificity. The conclusion on the effect of modification on specificity is questionable (see comments below). In general, it is a nice approach to combine ML to predict which combinational mutations will give better protein fitness. However, the prediction outcomes (precision, sensitivity, and specificity) are likely inflated by the data redundancy in the training and testing dataset. Most likely due to the residues selected, the degree of activity enhancement is still marginal. From the GUIDE-seq and mismatch results, the modification indicates a trends of increased cleavage at off-target sites as well. Thus, the SaCas9 variants does not possess an outstanding advantage.

Major comments

1. To address whether the MLDE can predict SpCas9 activity, the study used five different input sample size as testing: 5%, 10%, 20%, 50% and the 70%. Three random/diverse replicates were generated for this sampling size. For measuring the MLDE prediction outcome, the study used specificity, sensitivity, and precision to evaluate the prediction outcomes based on an arbitrary cutoff of 70%. It is unclear what are the testing dataset, the method section was not clear enough. Based on the Figure 1a, it seems that the full possible combinational dataset was used as test sets after the top three MLDE models were obtained. If this was the case, there will be an overlap between the training set and the testing data set, particularly when three random triplicates were used was training dataset. In this case, with the 10% dataset as input, the scale of input data size is closed to 30% for the MLDE model establishment.

In Figure 1, the variations for sample size with 50% and 70% is much smaller than the small sample size. One reason for this is that most of the data used for MLDE model training and testing are the same.

The authors should carefully address this issue ensuring that this high prediction outcome was not an artifact caused by data overlapping. One solution is to randomly subset 1/3 of the experiment data and only used that for testing the model.

2. When applying the MLDE for modeling the combinational mutations on SaCas9 efficiency, 300 randomly selected sites were used to establish the model. Similar to point 1, it is unclear whether the training dataset and the testing dataset is overlapped. From Figure 2, the study has highlighted the top variants with enriched scores among three tested gRNAs. For sg2, the model

predicts more variants with high enrichment score whilst there is not enrichment when measured experimentally. What is the cause that most variants from the MLDE are predicted as depleted? What is MLDE specificity, precision and sensitivity for the SaCas9 data?

3. There is a generally lack of statistics for the whole study. Figure 3b and 3c, marginal increase in GFP disruption (3b) and editing efficiency at endogenous sites (3c) by the SaCas9 variants were observed. However, it does not seem like that these increases are statistically significant. The author has excluded that expression levels of SaCas9 variant are similar in the cells. However, the current data is not convincingly enough to draw the conclusion. When defining the editing efficiency of the endogenous loci, T7E1 was used for quantifying indels. It should be noted that although T7E1 is conventional, the method is limited by its accuracy. This will make it more difficult to distinguish an activity difference by 10-30%. More accurate method for indel quantification should be used for this purpose.

4. Although the current study was not focus on the high fidelity SaCas9 and the results from 3c and 4a were not comparable, there is trends that KKH-SaCas9-SAV2 exhibits low activity compared to KKH-SaCas9.

5. It is plausible that the study has used GUIDE-seq to quantify the specificity of the SaCas9 variant. However, the current data does not support the claim that N88R-A889Q does not affect KKH-SaCas9 specificity. Figure 4c and S6, more off-target sites were found in the cells treated with the variant WT-plus. Figure S7, editing efficiency at the off-target sites are increased concordantly.

Other minor comments.

6. Abstract. KHH-SaCas9 should be KKH-SaCas9.

7. Figure S2, not quite sure if labeling to the b panel is correct or not. The stable report cell line should be GFP positive. The uninfected should be positive for GFP.

8. Figure S3, one mistake in the figure legend or the figure annotation. Left – right. Besides, it is unclear what that value of count means.

9. Figure S4, it is not clear whether this marginal increase is significant or not.

Reviewer #2:

Remarks to the Author:

In the manuscript "Machine learning-assisted engineering of activity-enhanced *Staphylococcus aureus* Cas9's KKH variant for genome editing", Thean et al. developed a machine learning-based Cas9 evolving system to optimize the KKH-SaCas9 and discovered several mutations within the WED and PI domains that enhance the editing efficiency of KKH-SaCas9. This method could be very useful to direct the engineering of new Cas9 and save a lot of bench work. But some aspects can be strengthened to improve the work.

First, for the first part, the authors tested the MLDE models for SpCas9. Dozens of Cas9 variants have been developed to either improve the editing fidelity or the editing scope. For validation, the authors can use the published data for MDLE models to predict certain type of Cas9 variants and compare the predicted mutations with the published ones.

Second, it's very important to perform significant analysis between Cas9 and the mutants in all the bar-graph figures, e.g Fig. 3b, 3c, 4c and supplementary figure panels. The authors should draw conclusions after significant analysis.

Third, more explanations are required to make the readers easier to follow the method. For example, a schematic representing the procedures and outcomes of each step should be placed in Figure 2; why choose those eight acid residues? Are they the only ones interacting with Cas9 PAM? How to calculate the total numbers of variants (e.g. a total of 1296 in the text)? Label errors in

Fig. S2a.

We sincerely thank both Reviewers for their detailed reading of the manuscript and their helpful and insightful suggestions. Based on the outstanding concerns of the Reviewers, we have performed additional experiments and analyses to enhance the quality of our manuscript. We believe that the substantial additional work incorporated into the revised manuscript has addressed all remaining issues, and hope that the Reviewers agree with us that the improved manuscript is now acceptable for publication in *Nature Communications*.

Reviewer #1:

The study by Dawn et al., reported the generation of activity-enhanced SaCas9 based on machine learning. The study used a machine learning package (MLDE) developed for predicting combinational protein fitness landscape to predict which combination of mutations can enhance Cas9 activity. The study first tested the MLDE with different sub-sets from a combinational mutation dataset (650 variants) of SpCas9 generated by the group in a previous study. The study found that with 10-20% of the variants as training dataset, the MLDE can predict functional variants with over 5-fold enrichment. The study then selected eight residues (three at the WED domain and five at the PI domains) at the KKH-SaCas9 protein. A total of 1296 combinational variants were designed. Combines MLDE prediction with 300 randomly picked data and experimental data generated by lentiviral library-based screening, the study showed that MLDE predicted SaCas9 variates with high activity is well correlated with those measured experimentally. The combined screening identified two variants exhibited marginal increased activity (see comments below), whereas the significance of increase was not documented yet. The study then applied structured-guided strategy to further improve the interaction of the two residues (N888 and A889) with residue in the PI domain, reasoning that increasing the interaction between WED domain the PI domain will increase the interaction between SaCas9 and the PAM. Four combined variants were tested, and one variant (N888R/A889Q) exhibited enhanced editing activity. Depending on the gRNAs tested, the activity was increased range from 11% to 30%, although whether these increases were significant or not remained to be clarified by the authors. And lastly, the study introduced the same mutations to an unpublished high-fidelity SaCas9 variant from the group. Similar degree of increased activity was observed. The study also applied GUIDE-seq to evaluate the off-target effects. Based on the results, the author concluded that the modification increases activity whilst does not compensate specificity. The conclusion on the effect of modification on specificity is questionable (see comments below). In general, it is a nice approach to combine ML to predict which combinational mutations will give better protein fitness. However, the prediction outcomes (precision, sensitivity, and specificity) are likely inflated by the data redundancy in the training and testing dataset. Most likely due to the residues selected, the degree of activity enhancement is still marginal. From the GUIDE-seq and mismatch results, the modification indicates a trends of increased cleavage at off-target sites as well. Thus, the SaCas9 variants does not possess an outstanding advantage.

We are grateful for the Reviewer's insightful suggestions. We also thank the Reviewer's comment that it is a nice approach to combine ML to predict which combinational mutations will give better protein fitness. Below please find specific responses to the Reviewer's concerns, and we believe our substantial additional work and analyses have improved the quality of our manuscript.

Here we would also like to supplement information as below to highlight the significance of our work. This study represents the first demonstration of integrating machine learning into large-scale experimental screening of multi-domain combinatorial mutagenesis to engineer the multi-dimensional activities of complex enzymes like the widely used CRISPR-Cas proteins. Our approach could change the current practices that rely solely on

either *in silico* prediction or a wet-lab-intensive approach towards combining both as a much more efficient strategy to engineer complex enzymes.

There was not a resource-efficient way to characterize a large spectrum of complex enzyme variants harboring multi-domain combinatorial mutations. Considering that any of the amino-acid sites of Cas9 enzyme in spatial proximity to the sgRNA-DNA complex are potential sites for optimization, which reaches over 40 sites spanning over its multiple domains, the number of combinatorial variants to screen through for optimization is too many (i.e., $2^{40} = 1.1 \times 10^{12}$) for wet-lab experiment even if each site is restricted to two (wild-type or mutated) amino-acid residues. Indeed, except for the datasets that we generated previously on characterizing several hundreds of SpCas9 variants *en masse*, other Cas9-engineering studies relied on methods including random and site-directed mutagenesis to select limited (often only tens of) clonal isolates for characterization. There is a lack of large-scale experimental screening datasets available/suitable for establishing machine learning-assisted approach for Cas9 engineering. The true discovery rate of variants with enhanced fitness/activity from an *in silico* screening could never be known without an experimental dataset. Experimental testing of only the selected top variants predicted is often used to decide how many among them result in actual increase in fitness or function. Even so, their activities are often compared to the wild type protein but not to the whole population of variants to see if they are truly top hits, not to mention the other top hits within the full mutational landscape could have been missed out. To understand how well machine learning can perform and decide if it can couple with experimental screening to reduce screening burden for identifying top variants, we have created new datasets for SaCas9 in this study. In our revised paper, we have now extended our analysis and cross-validated a total of 10 *in silico* and experimental datasets of multi-domain combinatorial mutagenesis libraries for Cas9 engineering, and we demonstrate for the first time that a machine learning-coupled combinatorial mutagenesis approach reduces the experimental screening burden by as high as 95% while accurately enriching top-performing Cas9 variants by ~7.5-fold compared to the null model.

In addition, we have now further demonstrated that our approach is capable of accurately predicting Cas9's activity in all three key aspects: 1) editing activity, 2) fidelity, and 3) targeting scope, and facilitate the identification of bona fide high-activity variants. The machine learning algorithm MLDE was previously applied to engineer other small-sized proteins with up to 4 mutations that lie in closer proximity in the protein sequence. Our study took a step forward in applying MLDE approach for a bigger and more complex genome editing enzyme (like Cas9) with up to 8 mutations that are scattered over the multiple domains of the protein, chosen by structure-guided mutagenesis. Even with the often greater difficulty with larger size and more substitutions of amino acid residues, our MLDE workflow gives high predictability and help reduce experimental screening burden by >80% to engineer Cas9's multi-dimensional activities for on- and off- target, PAM relaxation in parallel. For example, in our MLDE runs on SpCas9, using experimental screen data of only 130 (out of 952) variants for generating the training dataset allowed the identification of 17 top 5%-performing variants using MLDE, which represents a up to 3.8-fold increase in resource efficiency compared to a full-scale experimental screen (Please refer to Supplementary Table 8 and Discussion p.15-16 for details). High accuracy was also achieved when the same MLDE pipeline was applied to other cases to predict fitness for different sets of 5 to 8 residues at multiple domains being chosen for combinatorial mutagenesis. Our work also establishes parameters that can maximize MLDE usefulness in succeeding screens for accelerating Cas9 engineering with minimum wasted wet-lab resources.

In summary, we believe our approach for Cas9 engineering and our new SaCas9 variant datasets generated represent valuable strategy and resource that can be readily used in

many laboratories and industries for genome editor engineering and development of advanced machine learning methods for complex protein engineering. We believe that our work will appeal to the wide research community including but not limited to genome engineers, protein engineers, synthetic biologists, and computational biologists.

Major comments

1. To address whether the MLDE can predict SpCas9 activity, the study used five different input sample size as testing: 5%, 10%, 20%, 50% and the 70%. Three random/diverse replicates were generated for this sampling size. For measuring the MLDE prediction outcome, the study used specificity, sensitivity, and precision to evaluate the prediction outcomes based on an arbitrary cutoff of 70%. It is unclear what are the testing dataset, the method section was not clear enough. Based on the Figure 1a, it seems that the full possible combinational dataset was used as test sets after the top three MLDE models were obtained. If this was the case, there will be an overlap between the training set and the testing data set, particularly when three random triplicates were used as training dataset. In this case, with the 10% dataset as input, the scale of input data size is closed to 30% for the MLDE model establishment. In Figure 1, the variations for sample size with 50% and 70% is much smaller than the small sample size. One reason for this is that most of the data used for MLDE model training and testing are the same. The authors should carefully address this issue ensuring that this high prediction outcome was not an artifact caused by data overlapping. One solution is to randomly subset 1/3 of the experiment data and only used that for testing the model.

We sincerely thank the Reviewer's recommendations, and we apologize for insufficiency of the information provided. We have now updated our analysis to ensure the high MLDE prediction outcome was not affected by overlapping of training and testing data.

Specifically, for the Sg5 and Sg8 on-target datasets, we have now randomly subset and withheld 20% of variants in the library from the experimental dataset a priori, and they had never been fed to the MLDE algorithm. We used the dataset of editing activities measured from the CombiSEAL library of SpCas9 variants engineered at 8 amino-acid positions (R661, Q695, K848, E923, T924, Q926, K1003 and R1060) that interact with the sgRNA-DNA complex. First, we isolated 20% of the entire library (190 SpCas9 variants) as test data; among these selected variants, 122 of Sg5 and 136 of Sg8 have empirical measurements. We then generated input training datasets that do not overlap with the test data. The training datasets consist of 5%, 10%, 20%, 50%, and 70% of randomly drawn empirical measurements to test the minimal input for effective selection of top variants from MLDE prediction, corresponding to datasets of 33, 65, 130, 325, and 445 empirically measured Sg5 on-target activity and 37, 73, 146, 365, 510 Sg8 on-target activity measurements. We generated three replicates for each size, subjected to either randomized or diverse selection schemes for variants. To generate the randomized dataset, we used the `sample_n()` function from `dplyr` in R to randomly select the pre-defined number of E-scores. Taking the above-mentioned 20% (instead of 30%) of the entire library as non-overlapping test data allows the 70% randomly selected data not being the same for all three replicates, while maximizing the variant numbers for evaluation. With the above training and testing datasets, we have confirmed that MLDE using diverse and randomized training data led to similar high prediction results (Supplementary Figures 2; 3).

In our revised manuscript, we have further included new datasets to evaluate the MLDE prediction outcome for engineering Cas9's editing fidelity and editing scope. First, for editing fidelity, using the same method as for preparing the Sg5 on-target dataset, we prepared the datasets of Sg5 off-target activities (from Choi et al, Nature Methods, 2019) for MLDE. We withheld 190 variants as the test set for the off-target activities. Then we

randomly sampled the remaining variants that consist of 5%, 10%, 20%, 50%, and 70% of the library to generate 3 replicates of training datasets for each size, corresponding to 41, 83, 165, 414, and 579 empirical measurements of Sg5 off-target activity. The off-target activity was derived from min-max normalised E-score after setting a lower bound of -2.5. For Sg5 datasets, the on-target activity was measured in screens where the sgRNA and target site have perfectly matched protospacer sequence, while off-target activity was measured where the sgRNA is targeting a site that bear an artificially introduced synonymous mutation. Second, for editing scope, we used the dataset with a total of 58 SpCas9 variants bearing rational substitutions at five positions located in the PI domain that had their activities on non-canonical NGN PAMs assessed by HT-PAMDA (Walton et al., Science 2020). The on-target activity of the variant against 4 sgRNAs representing NGAT, NGCC, NGGG, and NGTA PAMs were used in the training data. To avoid having too few variants in the test set given the small dataset size, we withheld 29 variants (50% of the library) as test data that had not been fed to the MLDE algorithm for training, and performed MLDE with combinations of Bepler and Georgiev and modelling parameter p1 and p2 to predict on-target activity predictions using 10, 20, 25 and 50% input (empirical data of 29 and 15 variants).

We have now included the above details on the datasets used in our Methods section (p.21-23) and in our revised manuscript (p.7).

To confirm the high performance of MLDE prediction, we have performed evaluation with enrichment and normalized discounted cumulative gain (NDCG), which reflect the likelihood of identifying top-performing variants. NDCG, which compares the predicted ranking to the actual ranking, aligns with the goal of MLDE to identify high-fitness variants as top-ranking variants (Wu et al., PNAS, 2019; Wittmann et al., Cell Systems, 2021). If the predicted ranking and the actual ranking are identical, NDCG reaches its maximum value of 1. Models that misidentify low-fitness variants as top-ranking ones would result in low NDCG. Similarly, enrichment evaluates the likelihood of identifying the high-fitness variants among the top 5% hits predicted by the model compared to random selection. Enrichment provides us with an estimate of identifying high-fitness variants when we select the top 5% variants by predicted fitness for downstream experimental validation. When the larger fraction of highest-fitness variants is captured in the top 5% prediction, enrichment increases from 1. In our ML runs, we found that NDCG and enrichments were robust metrics for scoring models and parameter performances (Supplementary Figure 2), especially for Sg8 on-target activity where only about 10 variants (1.15% of the library) show activities comparable to wild type (Choi et al., Nature Methods, 2019). NDCG and enrichment were thus used for subsequent scoring, which align with our objectives to isolate the top-performing Cas9 variants. Looking into NDCG and enrichment, all the embeddings and models combinations performed well, while Bepler and Georgiev embeddings with p2 parameter outperformed other parameters when 5-20% of training data was fed to MLDE (Figure 1a).

Taking NDCG and enrichment together into consideration, we determined that 20% of input can be used as the input threshold that gave relatively robust and consistent performance in identifying top-performing candidates (Figure 1a, b; Supplementary Figure 4). 10% of input can also be used to further reduce the experimental screening burden with the metric scores slightly compromised (Figure 1a). Using merely 10% of input was sufficient to identify clusters of variants with high activity for the Sg5 dataset, and consistent identification of variants with at least 70% of wild-type activity across 10%, 20%, 50%, and 70% of input was observed (Supplementary Figure 3). MLDE runs on the Sg8 dataset again successfully identified the top-performing variants (Figure 1b, Supplementary Figure 5), albeit that NDCG and enrichment were lower than those observed for the Sg5 dataset (Figure 1a; see Supplementary Text). The top hits predicted from Sg5 and Sg8 datasets included

Opti-SpCas9 that was experimentally confirmed in our previous study to exhibit high on-target activities for both Sg5 and Sg8 (Choi et al., Nature Methods, 2019). Using MLDE, the enrichment in identifying top-performing variants reached about 8.6-fold for Sg5 (and about 5.8-fold for Sg8) with 20% input compared to the null model (Figure 1a; Supplementary Table 2). The enrichment reached about 7.5-fold with 5% and 10% input for Sg5 (Figure 1a; Supplementary Table 2). We further applied MLDE for off-target prediction. We took the same set of variants used for on-target activity prediction constituting 10, 20, 50 and 70% of empirical data of Sg5 off-target activities as training data for MLDE. MLDE achieved similarly high NDCG scores and about 5.5-fold enrichment with 20% input in off-target activity prediction (Supplementary Figure 6; Supplementary Table 3).

PAM relaxation is another key research area on SpCas9 engineering and thus we explored whether MLDE could facilitate screening on variants that cleave effectively on non-canonical PAMs. Specifically, we tested MLDE on SpCas9 variants' activities on non-canonical NGN PAMs from the abovementioned HT-PAMDA experiment. We run MLDE using 10, 20, 25 and 50% input (6, 12, 15, 29 variants). Due to the small size of the library, we were not able to calculate enrichment since there were only 3 variants warranted to be top 5% in the dataset. We focused on NDCG and again observed high scores on MLDE's prediction (Figure 1c; Supplementary Table 4). All the modelling parameters performed well when supplied with 50% training data while Bepler and Georgiev embeddings with p2 parameter outperformed other parameters when only 10 and 20% training data were fed to MLDE (Figure 1c). Across the four PAMs tested, supplying 20% training data to MLDE could achieve comparable performance to MLDE runs using 25 and 50% training data. Thus, we used MLDE results from 20% training data for the rest of the analysis. Looking into the best runs for each PAM, SpG was detected correctly to be amongst the top 20% variants with high activity at NGAT and NGCC PAMs (Figure 1d; Supplementary Figure 7).

Taking together the accurate prediction and the ability to isolate bona fide high-activity variants, we found that MLDE is compatible with rational-design guided library in various aspects of SpCas9 engineering.

We have also now included these new analyses in our revised manuscript (p.7-9; p.24-25). We hope that our detailed clarifications and revisions have addressed the Reviewer's concerns.

2. When applying the MLDE for modeling the combinational mutations on SaCas9 efficiency, 300 randomly selected sites were used to establish the model. Similar to point 1, it is unclear whether the training dataset and the testing dataset is overlapped. From Figure 2, the study has highlighted the top variants with enriched scores among three tested gRNAs. For sg2, the model predicts more variants with high enrichment score whilst there is not enrichment when measured experimentally. What is the cause that most variants from the MLDE are predicted as depleted? What is MLDE specificity, precision and sensitivity for the SaCas9 data?

We thank the Reviewer's comments, and we apologize for the insufficiency of the above information.

We have now added detailed description on our dataset generation for MLDE in the Methods section and have ensured that the MLDE prediction outcome was not affected by overlapping of training and testing data. The in-house SaCas9 dataset consists of 1,296 variants that were constructed and tested in this study. Substitutions on 8 amino acid positions (887, 888, 889, 985, 986, 988, 989, 991) that are widely scattered over the WED and PI domains were rationally chosen based on protein structure analyses (see Supplementary Table 3 for details). The SaCas9 variants' on-target activities against sg1, sg2,

and sg3 were measured as the E-score derived from the high-throughput fluorescent protein disruption assay. We again withheld 20% of the empirical data (260 variants) as the test set, and they have been unseen to MLDE algorithm. From the remaining variants, we generated 3 replicates of randomly selected datasets that consisted of 65, 130, 260, 648 and 907 variants that corresponding to 5, 10, 20, 50, and 70% of the full library as training data for MLDE.

We also thank the Reviewer for raising the point that the model predicts more variants with high enrichment score whilst there is not enrichment when measured experimentally in some cases, which could be due to the embedding and modelling parameters used. We run MLDE using the training data of different sizes and evaluated the MLDE performance using the test set variants, and we have compared our *in silico* prediction results and experimental screen data. MLDE using the Georgiev embedding with the ensemble of random forest and SVM algorithm (parameter 2) (i.e., Georgiev.p2) showed the best performance (revised Figure 3a, Supplementary Table 7). The enrichment in identifying top-performing variants reached about 6.7-fold for sg1, 9.2-fold for sg2, 5.1-fold for sg3 with 20% input, and about 5.1-fold for sg1, 7.2-fold for sg2, 4.1-fold for sg3 with 10% input performance (revised Figure 3a, Supplementary Table 7). Although using the other parameters (i.e., Bepler.p1, Bepler.p2, and Georgiev.p1) also achieved high enrichment scores (Supplementary Figures 9-11), our results indicated that these parameters gave more predicted variants with high enrichment score that were not enriched in the experimental datasets. Indeed, Georgiev.p2 parameter gave the best prediction performance across most datasets used in our ten *in silico* and experimental cross-validation work throughout this study. Our findings also indicate the importance of this work to help select the best-performing embedding and modelling parameters for more consistent predictions in succeeding screens.

In addition, we noticed that for certain datasets that lack high-fitness variants in the training input could result in most variants from MLDE being predicted as depleted. Specifically, when training datasets only contained variants with poor activities (< 55% of the activity of the top experimentally validated variant), MLDE performance was hindered (Supplementary Figures 9-11). Such condition was prominent in datasets of sg3 that 2 out of 3 training datasets failed to sample any high-fitness variants. Our results are in line with the MLDE developer's recommendation that we ought to focus on surveying diverse sequence spaces believed to contain functional variants for MLDE (Wittmann et al., Cell Systems, 2021). Thus, the good performance of MLDE also requires the presence of variants with higher fitness in the input training datasets. Our results thus highlight the importance of improving strategies on sampling more variants with higher fitness to yield high-quality MLDE predictions.

Overall, with datasets that contained higher fitness variants for our MLDE runs, we found that the three independent sets of activity measurements on KKH-SaCas9 variants using sgRNA sg1, sg2 or sg3 yielded consistent predictions with the experimental screen data, especially in MLDE predictions using the Georgiev embedding and modelling parameter 2 (Figure 3b; Supplementary Figure 8). This result is in line with our SpCas9 activity prediction showing that MLDE identifies top-performing variants readily. The top-5%-hits predicted from the three sets included N888Q and N888Q/A889S variants identified in our experimental screen data (Figure 3b). The high level of consistency, in particular the identification of the common top-performing variants, between the *in silico* and experimental screen data confirms that the MLDE model can be used to predict KKH-SaCas9's variants with high activity. In addition to NDCG and enrichment, we have included specificity, precision, and sensitivity metrics in Supplementary Table 7 to present the MLDE performance for the SaCas9 data.

We have now included the above analyses in our revised manuscript (p.11-12; p.23). We hope that our detailed clarifications and revisions have addressed the Reviewer's concerns.

3. There is a generally lack of statistics for the whole study. Figure 3b and 3c, marginal increase in GFP disruption (3b) and editing efficiency at endogenous sites (3c) by the SaCas9 variants were observed. However, it does not seem like that these increases are statistically significant. The author has excluded that expression levels of SaCas9 variant are similar in the cells. However, the current data is not convincingly enough to draw the conclusion. When defining the editing efficiency of the endogenous loci, T7E1 was used for quantifying indels. It should be noted that although T7E1 is conventional, the method is limited by its accuracy. This will make it more difficult to distinguish an activity difference by 10-30%. More accurate method for indel quantification should be used for this purpose.

We sincerely thank the Reviewer's comments and suggestions, and we apologize for the missing information. We have added back statistical analyses to confirm the significant increases in GFP disruption (revised Figure 4b) and editing efficiency at endogenous sites (revised Figure 4c) by KKH-SaCas9-plus are statistically significant. We have now also performed deep sequencing assay, a more accurate method for indel quantification, to confirm the significant enhancement (17-33%) of KKH-SaCas9-plus's editing efficiency at the endogenous sites (Figure 4d). To strengthen our conclusion that our identified mutations could increase KKH-SaCas9's editing activity, we have extended our validation work by grafting the N888R/A889Q mutations onto the KKH-SaCas9-derived cytosine base editor (BE4max). Our deep sequencing results found that these mutations also increased the base editor's activity at four endogenous loci (by 11-93% at the most edited base within the target sites) (Supplementary Figure 14). This result suggests that the increased editing activity brought by the mutations is likely dictated at the DNA binding level. We have now included these new data and analyses in our revised manuscript (p.14).

4. Although the current study was not focus on the high fidelity SaCas9 and the results from 3c and 4a were not comparable, there is trends that KKH-SaCas9-SAV2 exhibits low activity compared to KKH-SaCas9.

We thank the Reviewer for raising this point. In the work on characterizing KKH-SaCas9-SAV2 (Yuen et al., *Nucleic Acids Research*, 2022), it was noticed that this variant exhibited lower editing activity compared to KKH-SaCas9 while acquiring increased ability to discriminate single-base mismatches. This is in line with the results from revised Figure 4c and Supplementary Figure 15c. These results indicate the need to increase KKH-SaCas9-SAV2's editing activity. We thus tested whether the addition of N888R/A889Q could improve the activity of KKH-SaCas9-SAV2. We found that N888R/A889Q also enhanced the on-target activity of SAV2 (i.e., showed 121% of SAV2's activity, averaged from sgRNAs targeting 8 loci), while 5 out of the 8 loci showed 9-48% enhancement of the editing activity (revised Supplementary Figure 15c, d). This combined mutant (KKH-SaCas9-SAV2-plus) generated comparably few genome-wide off-target edits (revised Supplementary Figure 15a, b), while we observed increased edits at some of the tested off-target sites with single mismatches (revised Supplementary Figure 16). Our results indicate the feasibility to combine activity- and specificity- enhancing mutations for further optimizing the KKH-SaCas9's performance. This data also affirms that the abilities of KKH-SaCas9 to bind the DNA and distinguish base mismatches between sgRNA and the DNA target probably act

through distinct mechanisms, and thus its activity and specificity could be engineered independently. We have now clarified these points in our revised manuscript (p.17).

5. It is plausible that the study has used GUIDE-seq to quantify the specificity of the SaCas9 variant. However, the current data does not support the claim that N888R-A889Q does not affect KKH-SaCas9 specificity. Figure 4c and S6, more off-target sites were found in the cells treated with the variant WT-plus. Figure S7, editing efficiency at the off-target sites are increased concordantly.

We thank the Reviewer's comment, and we apologize for the lack of clarification on this point. We acknowledge that while N888R/A889Q increases the on-target activity of KKH-SaCas9, it may also increase off-target editing. Our GUIDE-seq results indicated that KKH-SaCas9-plus showed comparable on-to-off target editing ratio to wild-type, albeit that there were alternative off-target sites identified (revised Supplementary Figure 15a, b). KKH-SaCas9-plus showed more off-target edits at some of the target sequences with single-base mismatches (revised Supplementary Figure 16). We have now elaborated our data interpretation and updated our claim in our revised manuscript (p.17).

Other minor comments.

6. Abstract. KHH-SaCas9 should be KKH-SaCas9.

We apologize for the typo and have made the correction in our revised manuscript.

7. Figure S2, not quite sure if labeling to the b panel is correct or not. The stable report cell line should be GFP positive. The uninfected should be positive for GFP.

We thank the Reviewer for raising this point. We have now updated our labels to clarify on which samples harbor or not the reporter, KKH-SaCas9, and sgRNA in our revised Figure 2b.

8. Figure S3, one mistake in the figure legend or the figure annotation. Left – right. Besides, it is unclear what that value of count means.

We apologize for the mistake and missing information. We have now corrected the figure legend of our revised Supplementary Figure 8 to indicate the top and bottom panels refer to the experimental screen data and MLDE prediction, respectively. We have also now indicated in the figure legend that the value of count means the occurrences of the amino-acid residues per site among the top 5% variants identified in the experimental screens and the best MLDE runs using Georgiev embedding and modelling parameter 2.

9. Figure S4, it is not clear whether this marginal increase is significant or not.

We thank the Reviewer for raising this point. We have now carried out statistical analyses to confirm the significant increase in GFP disruption brought by the N888Q variant (revised Supplementary Figure 12). The validation results were consistent with the screening data, from which we revealed that the N888Q variant exhibited increased editing activities over KKH-SaCas9 when paired with sg1 and sg3 sgRNAs.

Reviewer #2:

In the manuscript “Machine learning-assisted engineering of activity-enhanced *Staphylococcus aureus* Cas9’s KKH variant for genome editing”, Thean et al. developed a machine learning-based Cas9 evolving system to optimize the KHH-SaCas9 and discovered several mutations within the WED and PI domains that enhance the editing efficiency of KHH-SaCas9. This method could be very useful to direct the engineering of new Cas9 and save a lot of bench work. But some aspects can be strengthened to improve the work.

We are grateful for the Reviewer’s support of our paper and insightful suggestions. We also thank the Reviewer’s comment that our method could be very useful to direct the engineering of new Cas9 and save a lot of bench work. Below please find specific responses to the reviewer’s remaining concerns, which we believe have improved the quality of the work.

1. First, for the first part, the authors tested the MLDE models for SpCas9. Dozens of Cas9 variants have been developed to either improve the editing fidelity or the editing scope. For validation, the authors can use the published data for MDLE models to predict certain type of Cas9 variants and compare the predicted mutations with the published ones.

We sincerely thank the Reviewer’s recommendations. We have now analyzed additional published datasets and validated the good MLDE prediction outcome for engineering Cas9’s editing fidelity and editing scope.

First, we have now applied MLDE for off-target prediction. We took the same set of variants used for on-target activity prediction constituting 10, 20, 50 and 70% of empirical data of Sg5 off-target activities from Choi et al., Nature Methods, 2019 as training data for MLDE. We evaluated MLDE prediction performance with enrichment and normalized discounted cumulative gain (NDCG), which reflect the likelihood of identifying top-performing variants. NDCG, which compares the predicted ranking to the actual ranking, aligns with the goal of MLDE to identify high-fitness variants as top-ranking variants (Wu et al., PNAS, 2019; Wittmann et al., Cell Systems, 2021). If the predicted ranking and the actual ranking are identical, NDCG reaches its maximum value of 1. Models that misidentify low-fitness variants as top-ranking ones would result in low NDCG. Similarly, enrichment evaluates the likelihood of identifying the high-fitness variants among the top 5% hits predicted by the model compared to random selection. Enrichment provides us with an estimate of identifying high-fitness variants when we select the top 5% variants by predicted fitness for downstream experimental validation. When the larger fraction of highest-fitness variants is captured in the top 5% prediction, enrichment increases from 1. In our runs, we found that MLDE achieved high NDCG scores for Sg5 off-target prediction (which is similar to those achieved for Sg5 on-target prediction) and about 5.5-fold enrichment with 20% input in off-target activity prediction (Supplementary Figure 6; Supplementary Table 3).

Second, PAM relaxation is another key research area on SpCas9 engineering and thus we explored whether MLDE could facilitate screening on variants that cleave effectively on non-canonical PAMs. Specifically, we have now tested MLDE on SpCas9 variants’ activities on non-canonical NGN PAMs from the previously published High-Throughput PAM Determination Assay (HT-PAMDA) experiment (Walton et al., Science 2020). We run MLDE using 10, 20, 25 and 50% input (6, 12, 15, 29 variants). Due to the small size of the library, we were not able to calculate enrichment since there were only 3 variants warranted to be top 5% in the dataset. We looked at NDCG and again observed high scores on MLDE’s prediction (Figure 1c; Supplementary Table 4). All the modelling parameters performed well when supplied with 50% training data while Bepler and Georgiev embeddings with p2

parameter outperformed other parameters when only 10 and 20% training data were fed to MLDE (Figure 1c). Across the four PAMs tested, supplying 20% training data to MLDE could achieve comparable performance to MLDE runs using 25 and 50% training data. Thus, we used MLDE results from 20% training data for the rest of the analysis. Looking into the best runs for each PAM, the previously validated SpG variant was detected correctly to be amongst the top 20% variants with high activity at NGAT and NGCC PAMs (Figure 1d; Supplementary Figure 7).

Despite dozens of Cas9 variants that have been developed to either improve the editing fidelity or scope, most of those Cas9-engineering studies relied on methods including random and site-directed mutagenesis to select limited (often less than or only tens of) clonal isolates for characterization. Thus, there are only limited large-scale experimental screening datasets available/suitable for MLDE validation.

In sum, we have successfully applied MLDE for the first time to predict Cas9's activity in three key aspects: editing activity, fidelity, and targeting scope. We have now included these new validation analyses in our revised manuscript (p.7-9; p.15) and the Methods section (p.21-25).

2. Second, it's very important to perform significant analysis between Cas9 and the mutants in all the bar-graph figures, e.g Fig. 3b, 3c, 4c and supplementary figure panels. The authors should draw conclusions after significant analysis.

We thank the Reviewer for raising this point, and we apologize for the missing information. We have added back statistical analyses in the bar-graph figures to confirm the significant increases in GFP disruption (revised Figure 4b and Supplementary Figure 12) and editing efficiency at endogenous sites (revised Figure 4c) by KKH-SaCas9 variants are statistically significant. We have further performed deep sequencing assay, a more accurate method for indel quantification, and confirmed the significant enhancement (17-33%) of KKH-SaCas9-plus's editing efficiency at the endogenous sites (revised Figure 4d). With the statistical analyses, we have now confirmed that N888R/A889Q also significantly enhanced the on-target editing activity of SAV2 at the endogenous loci (revised Supplementary Figure 15c, d). To strengthen our conclusion that our identified mutations could increase KKH-SaCas9's editing activity, we have extended our validation work by grafting the N888R/A889Q mutations onto the KKH-SaCas9-derived cytosine base editor (BE4max). Our deep sequencing results found that these mutations also significantly increased the base editor's activity at four endogenous loci (revised Supplementary Figure 14). This result suggests that the increased editing activity brought by the mutations is likely dictated at the DNA binding level. We have now indicated the statistical analyses in the revised figures and legends, as well as updated the text based on conclusions drawn after significant analysis (p.14) in our revised manuscript.

3. Third, more explanations are required to make the readers easier to follow the method. For example, a schematic representing the procedures and outcomes of each step should be placed in Figure 2; why choose those eight acid residues? Are they the only ones interacting with Cas9 PAM? How to calculate the total numbers of variants (e.g. a total of 1296 in the text)? Label errors in Fig. S2a.

We thank the Reviewer for raising these points, and we appreciate the Reviewer's recommendations.

We have now included a schematic representing the procedures and outcomes of each step (revised Figure 5). We started with structure-guided design to select sites and residues

for mutagenesis and built multi-domain combinatorial variant libraries. We then run MLDE and tested embedding and model parameters to generate *in silico* predictions. With our cross-validation of a total of ten *in silico* and experimental datasets of multi-domain combinatorial mutagenesis libraries for Cas9 engineering, our work informed parameters for accurate prediction of Cas9's activity in three key aspects: 1) editing activity, 2) fidelity, and 3) targeting scope, and the identification of true high-performing variants. We show that integrating machine learning into large-scale experimental screening of multi-domain combinatorial mutagenesis reduces experimental screening burden by as high as 95% while enriching top-performing Cas9 variants by ~7.5-fold compared to the null model. This approach also increases the resource efficiency by up to 3.8-fold compared to a full-scale experimental screen (details presented in Supplementary Table 8 and Discussion p.15).

We have now also added more details to explain our selection of the eight amino acid residues for library variant generation. We sought to augment the editing activity of KKH-SaCas9 and speculated that introducing additional non-base-specific interactions between KKH-SaCas9 and the PAM duplex of the target DNA could increase the enzyme's efficiency. Such strategy was shown effective in compensating the reduced DNA base-specific interactions of an engineered SpCas9 variant that broaden its PAM compatibility and restoring the enzyme's activity (Nishimasu et al., Science, 2018). For SaCas9, Nishimasu et al., Cell 2015 has illustrated in the crystal structure (5CZZ) its amino acid residues that show direct contact with the target DNA of the PAM duplex. Specifically, it was highlighted that amino acid residues at position 985, 986, 991, and 1015 on its PI domain form water-mediated hydrogen-bonds with the non-target DNA strand at the PAM duplex, while residues at positions 789, 882, 886, 888, 889, and 909 on its WED domain interact with the phosphate backbone of the PAM duplex. Mutations at positions 988 and 989 were also reported to alter SaCas9's PAM constraint (Ma et al., Nature Communications, 2019). In this study, we focused on modifying eight amino acid positions (887, 888, 889, 985, 986, 988, 989, and 991) that interact with and surround the PAM duplex for combinatorial mutagenesis (Figure 2a; Supplementary Table 5). Up to two amino-acid alternatives to the wild-type residue were selected for each site based on structural predictions. This could potentially increase non-base-specific interactions between KKH-SaCas9 and the DNA and relieve the PAM constraint. To facilitate the changes, we selectively chose sites in the WED domain to reinforce the protein binding to the DNA backbone (Supplementary Table 5). This led to a total of 1,296 variant combinations including the wild-type residues (i.e., 12 mutation combinations at WED domain x 108 mutation combinations at PI domain). We did not modify residue position 1015 because this R1015H was shown to be important for maintaining the high activity of KKH-SaCas9 to act on NNNRRT PAM (Kleinstiver et al., Nature Biotechnology, 2015). Residue positions 789, 882, 886, and 909 were not included to confine the library size for combinatorial mutagenesis, while they are potential sites for future engineering. We have included the above information in our revised manuscript (p.9-10) and revised Figure 2a.

We apologize for the label errors, and we have now made the corrections to clarify on which samples harbor or not the reporter, KKH-SaCas9, and sgRNA in our revised Figure 2b.

Reviewers' Comments:

Reviewer #1:

Remarks to the Author:

In this revision, they have performed substantial analyses to support the validity of the ML-based prediction of SaCas9 variants with increased activity. Targeted deep sequencing was performed to validate the increased activities in some gRNA sites. They have also showed that the activity increased variant also increase base editing efficiency. The study provides a nice ML framework for the prediction of Cas9 variants with enhanced properties, in this case with on-target activity. However, I still have to point out that the level of activity enhancement with KKH-SaCas9-plus, as compared to KKH-SaCas9, is still quite low (17-33%) in 2 out 3 gRNAs (Figure 2) and 3 out of 7 gRNAs (Figure 4) tested. In the discussion (p.17, line 411-413) the authors have commented that analyzing a large panel of sgRNA pairs is required to define the design rules of KKH-SaCas9-plus. Apart from this, the authors have thoroughly addressed all my concerns in the previous version of their manuscript. A few minor comments are highlighted below for consideration of changes, not very critical:

1. abstract: line 33 "one of the variants", better write out the mutant (N888R/A889Q).
2. abstract: line 34 "...at multiple endogenous loci...", please write the exact number, "multiple is not precise". It is important to highlight here, how many gRNAs out of the total number of gRNAs tested exhibited enhance activity.
3. Line 53, "variants", better changed to "orthologs".
4. Line 96, "... the WED domain of SaCas9 thus far", include a short introduction to the WED domain function, i.e., "which is responsible for the gRNA scaffold recognition".
5. Line 121, "; we", should be ". We"
6. Line 214, is 887 missing in the sentence? "..., while residues at position 789, 882, 886, 888, and 909..."
7. Discussion, line 445, I will be careful to pursuing the concept of "PAMless". This will mean that delivering the PAMless Cas9 with e.g., plasmids and DNA viral vectors will encounter the problem of self-targeting.
8. Data availability, there is no accession number for the GUIDE-seq, or targeted deep sequencing data.
9. Figure 2 A, color for the Bin A. It might be better to use blue color instead of red. I first thought that it was referring to RFP. Figure 2d, scale is needed for the enrichment score of sg2.

Reviewer #2:

Remarks to the Author:

The authors have addressed all my concerns. The revised manuscript has been greatly improved and is clear to me. Therefore, I recommend the publication of this work in Nature Communications.

We sincerely thank all Reviewers for their detailed reading of the manuscript and their positive feedbacks, and the editors for offering in principle our manuscript to be published in *Nature Communications*. Based on the editorial guidelines for the final submission of our manuscript, we have completed the required documents and revisions. Below please also find our responses to the Reviewers' comments. We look forward to receiving the formal acceptance of our manuscript and its publication in *Nature Communications*.

Reviewer #1:

In this revision, they have performed substantial analyses to support the validity of the ML-based prediction of SaCas9 variants with increased activity. Targeted deep sequencing was performed to validate the increased activities in some gRNA sites. They have also showed that the activity increased variant also increase base editing efficiency. The study provides a nice ML framework for the prediction of Cas9 variants with enhanced properties, in this case with on-target activity. However, I still have to point out that the level of activity enhancement with KKH-SaCas9-plus, as compared to KKH-SaCas9, is still quite low (17-33%) in 2 out of 3 gRNAs (Figure 2) and 3 out of 7 gRNAs (Figure 4) tested. In the discussion (p.17, line 411-413) the authors have commented that analyzing a large panel of sgRNA pairs is required to define the design rules of KKH-SaCas9-plus. Apart from this, the authors have thoroughly addressed all my concerns in the previous version of their manuscript. A few minor comments are highlighted below for consideration of changes, not very critical:

We are grateful that the Reviewer found all the concerns in the previous version of the manuscript were thoroughly addressed, and our work provides a nice ML framework for the prediction of Cas9 variants with enhanced properties, in this case with on-target activity. We acknowledge that the level of activity enhancement with KKH-SaCas9-plus shown is low (17-33%) in 2 out of 3 gRNAs (Figure 2) and 3 out of 7 sgRNAs (Figure 4) tested. As also pointed out by the Reviewer, we have added comments in the discussion that analyzing a large panel of sgRNA pairs is required to define the design rules of KKH-SaCas9-plus in future. Below we have addressed the remaining minor comments raised by the Reviewer. Here we would like to thank again the Reviewer for his/her insightful comments throughout the peer-review process.

1. abstract: line 33 "one of the variants", better write out the mutant (N888R/A889Q).

We appreciate the Reviewer's suggestion. "N888R/A889Q" mutant is now written out in the abstract.

2. abstract: line 34 "...at multiple endogenous loci...", please write the exact number, "multiple is not precise". It is important to highlight here, how many gRNAs out of the total number of gRNAs tested exhibited enhance activity.

We appreciate the Reviewer's suggestion and agree that it would be precise to write the exact number. However, due to word limitation, we have to shorten our abstract and the technical details on the editing performance of the variant are now removed from the abstract. The exact number has been described in p.14 of our manuscript.

3. Line 53, "variants", better changed to "orthologs".

We thank the Reviewer's suggestion. "variants" is now changed to "orthologs".

4. Line 96, "... the WED domain of SaCas9 thus far", include a short introduction to the WED domain function, i.e., "which is responsible for the gRNA scaffold recognition".

We thank the Reviewer's suggestion. We have now added a short introduction (i.e., "which is responsible for the gRNA scaffold recognition" to describe the WED domain function in the sentence.

5. Line 121, "; we", should be ". We"

We thank the Reviewer's suggestion. We have now updated "; we" as ". We".

6. Line 214, is 887 missing in the sentence? "..., while residues at position 789, 882, 886, 888, and 909..."

We apologize for missing 887 in the sentence. We have now added it back in the sentence.

7. Discussion, line 445, I will be careful to pursuing the concept of "PAMless". This will mean that delivering the PAMless Cas9 with e.g., plasmids and DNA viral vectors will encounter the problem of self-targeting.

We thank the Reviewer for raising this point. We have now replaced "PAMless SaCas9" with "SaCas9 with much relaxed PAM constraint" in the sentence.

8. Data availability, there is no accession number for the GUIDE-seq, or targeted deep sequencing data.

We thank the Reviewer for raising this point. We have now added the accession numbers for the GUIDE-seq and deep sequencing data.

9. Figure 2 A, color for the Bin A. It might be better to use blue color instead of red. I first thought that it was referring to RFP. Figure 2d, scale is needed for the enrichment score of sg2.

We thank the Reviewer for raising this point and we apologize for the confusion. We have updated Figure 2A to use blue color instead of red for the Bin A. We have also added the scale for the enrichment score of sg2 in Figure 2d.

Reviewer #2:

The authors have addressed all my concerns. The revised manuscript has been greatly improved and is clear to me. Therefore, I recommend the publication of this work in Nature Communications.

We are grateful that the Reviewer found all the concerns were addressed and the revised paper is greatly improved and clear, as well as recommends the publication of this work in *Nature Communications*. We would like to thank again the Reviewer for his/her insightful comments throughout the peer-review process.